# Out-of-the-box calving front detection method using deep learning

Oskar Herrmann[1], Nora Gourmelon[2], Thorsten Seehaus[1], Andreas Maier[2], Johannes J. Fürst[1], Matthias H. Braun[1], and Vincent Christlein[2]

[1]Institute of Geography, Friedrich-Alexander-Universität Erlangen-Nürnberg, Erlangen, Germany
[2]Pattern Recognition Lab, Friedrich-Alexander-Universität Erlangen-Nürnberg, Erlangen, Germany

**Correspondence:** Oskar Herrmann (oskar.herrmann@fau.de)

**Abstract.** Glaciers across the globe react to the changing climate. Monitoring the transformation of glaciers is essential for projecting their contribution to global mean sea level (GMSL) rise. The delineation of glacier-calving fronts is an important part of the satellite-based monitoring process. This work presents a calving front extraction method based on the deep learning framework nnU-Net, which stands for no new U-Net. The framework automates the training of a popular neural network, called U-Net, designed for segmentation tasks. Our presented method marks the calving front in Synthetic Aperture Radar (SAR) images of glaciers. The images are taken by six different sensor systems. A benchmark dataset for calving front extraction is used for training and evaluation. The dataset contains two labels for each image. One label denotes a classic image segmentation into different zones (glacier, ocean, rock, and no information available). The other label marks the edge between the glacier and the ocean, i. e., the calving front. In this work, the nnU-Net is modified to predict both labels simultaneously. In the field of machine learning, the prediction of multiple labels is referred to as Multi-Task-Learning (MTL). The resulting predictions of both labels benefit from simultaneous optimization. For further testing of the capabilities of MTL, two different network architectures are compared, and an additional task, the segmentation of the glacier outline, is added to the training. In the end, we show that fusing the label of the calving front and the zone label is the most efficient way to optimize both tasks with no significant accuracy reduction compared to the MTL neural network architectures. The automatic detection of the calving front with a nnU-Net trained on fused labels improves from the baseline Mean Distance Error (MDE) of $753 \pm 76\,\mathrm{m}$ to $541 \pm 84\,\mathrm{m}$. The scripts for our experiments are published on GitHub (https://github.com/ho11laqe/nnUNet_calvingfront_detection). An easy access version is published on Hugging Face (https://huggingface.co/spaces/ho11laqe/nnUNet_calvingfront_detection).

## 1 Introduction

Unlike the large majority of land-terminating glaciers, marine and lake-terminating (MALT) glaciers reach a water body at a low elevation. The contact surface is often referred to as the calving front. Their ice is lost by sub-marine melting or calving, i. e., ice that breaks off, gets de-connected, and starts to float freely in the form of icebergs or ice floes. Both processes, sub-marine melting and calving, determine the total frontal ablation, i. e., the mass loss at the calving front. Frontal ablation is often a dominant factor in the total mass budget of MALT glaciers (McNabb et al., 2015; Shepherd et al., 2018; Minowa et al., 2021). Besides its importance in the total glacier mass balance, the representation of processes controlling frontal ablation is currently a pressing task for numerical glacier models (Beer et al., 2021). Neglecting frontal ablation can introduce an important bias.

Recinos et al. (2019) analyzed the impact on ice thickness reconstruction (based on mass conservation) in Alaska and reported an underestimation of 19 % on regional and up to 30 % on glacier scales. Various successful approaches exist to parameterize frontal ablation for individual glaciers. Still, the implementation in large-scale or global models is limited by the amount and quality of measurements to constrain the models (Recinos et al., 2019). Thus, large-scale measurements (ideally time series) of frontal ablation are demanded by the modelling community (Recinos et al., 2021). Driving forces of frontal ablation are, on the one hand, ice flux (higher flux, e. g., due to bed lubrication by meltwater can trigger calving events) and, on the other hand, marine factors such as ocean temperature, fjord bathymetry, and sea ice/ice mélange conditions (Carr et al., 2014; Straneo et al., 2013). For example, the persistence of the ice mélange in front of the glacier can stabilize the calving front and affect the glacier dynamics. In contrast, the breakup of the ice mélange can lead to increased calving and ice flow at the glacier terminus (Amundson et al., 2010; Kneib-Walter et al., 2021; Rott et al., 2020). Moreover, a significant frontal retreat can also indicate a retreat of the grounding zone (Friedl et al., 2018). The retreat of the grounding zone of a glacier with retrograde bedrock formation will lead to further grounding zone retreat, resulting in increased ice loss and destabilization of the glacier or ice stream (Robel et al., 2016). Thus, information on the temporal variability of the calving front position provides fundamental information on the state of the glacier or ice stream. Therefore, the glacier area has been defined as an Essential Climate Variable (ECV) product by the World Meteorological Organization (WMO). Calving front positions were usually manually mapped using different remote sensing imagery (Baumhoer et al., 2018). Only a few studies applied automatic or semi-automatic approaches. In polar regions, the ocean downstream of the glaciers is often covered by sea ice and calved off icebergs, forming the so-called ice mélange, making calving front delineation a challenging task, even when captured by hand. Deep learning approaches have shown high potential for carrying out such complex segmentation tasks, e. g., on medical imagery (Jang and Cho, 2019). In recent years, the application of deep learning techniques for glacier front detection started (Zhang et al., 2019; Cheng et al., 2021; Baumhoer et al., 2019; Hartmann et al., 2021; Mohajerani et al., 2019; Baumhoer et al., 2021; Zhang et al., 2021; Heidler et al., 2021; Marochov et al., 2020; Loebel et al., 2022). Calving fronts can be located in both optical and SAR imagery. In optical imagery, calving fronts are more easily distinguishable, whereas SAR imagery has a higher scene availability, as it is independent of daytime, season and cloud coverage (Baumhoer et al., 2018). A direct comparison between the results of existing deep learning-based calving front extraction studies is not possible, as the models have been trained on different data, tested on different test sets, and evaluated using slightly differing metrics.

The benchmark dataset published by Gourmelon et al. (2022) provides 681 SAR images of calving fronts. SAR imagery is independent of sunlight and cloud coverage, enabling continuous temporal coverage of the observation area, but compared to optical data, it has only one channel and has more speckle noise. For every SAR image, two labels are provided. One label provides four classes: ocean, glacier, rock, and no information available (e. g., radar shadow and layover areas, areas outside the swath). The other label marks the calving front with a one-pixel wide line. Based on the training set of the dataset, Gourmelon et al. (2022) train a modified U-Net for each label. One U-Net solves the task of glacier segmentation, and one detects the calving front. On the two test glaciers of the dataset, the segmentation model achieves an Intersection over Union (IoU) of $67.7 \pm 0.6$. By taking the boundary between ocean and glacier, they extract a prediction of the calving front from the zone prediction with a Mean Distance Error (MDE) of $753 \pm 76$ m. The model trained directly on the front label achieved an MDE of $887 \pm 189$ m.

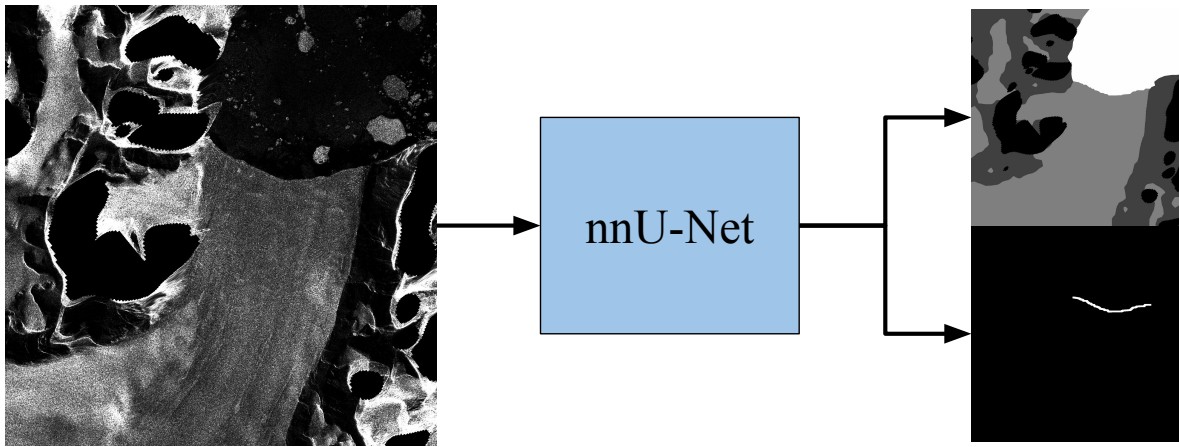

**Figure 1.** Illustration of the modified nnU-Net for simultaneous predicting landscape zones and the glacier front. On the left is an exemplary satellite image of the Crane Glacier taken by TanDEM-X (TDX) on 24 June 2011. On the right, the upper image shows the four segmentation classes: ocean (white), glacier (light grey), rock (dark grey), and no information available (black). The lower image shows the glacier's calving front as a white line versus the black background.

In this work, we present a method that utilises both labels of the dataset instead of training separate models for each task (see Fig. 1). Multi-Task-Learning (MTL) is a technique for machine learning algorithms that uses one model to tackle multiple tasks. In most cases, the potentially larger dataset and higher information content lead to higher performance for the individual tasks (Bischke et al., 2019; He et al., 2021; Li et al., 2019; Amyar et al., 2020; Chen et al., 2019).

Our method is based on the nnU-Net (no new U-Net) proposed by Isensee et al. (2021), which is an out-of-the-box framework for training the U-Net. The framework contains proven deep learning techniques and established hyperparameter values, as well as rule-based parameters that depend on the properties of the dataset and available GPU-memory. Therefore, the nnU-Net is a powerful tool that simplifies the application of deep learning algorithms. Its performance on segmentation of SAR data has not been tested, and the availability of two labels suggests the modification of the nnU-Net for MTL. As a baseline for our

evaluation of the nnU-Net, we use the results of Gourmelon et al. (2022).

In particular, our contributions are as follows:

(1) Visualizations showing the temporal and spatial distribution of the dataset by Gourmelon et al. (2022). (2) Application and evaluation of the nnU-Net for calving front detection and zone segmentation. (3) Two different modifications of the original nnU-Net to incorporate both labels for multi-task learning. (4) Test if an artificial third label improves the calving front detection

(5) Introduction of an efficient approach that fuses the two labels of the dataset. (6) Analysis of the influence of season, glacier, and satellite on the performance of the model.

The paper is organized as follows: After presenting the related work in Sec. 2, we give an overview of the dataset (Gourmelon et al., 2022) in Sec. 3. Section 4 explains the method and our six experimental setups. Section 5 examines the results and analyses the influence of different properties of the satellite images. Section 6 summarises the work.

## 2 Related Work

Automated monitoring of glacier-covered areas is a growing research field. Recent glacier monitoring uses deep learning methods due to the increasing availability of satellite images and computing power. Many methods are based on the U-Net (Ronneberger et al., 2015). They modify the vanilla U-Net for better performance on calving front detection (Loebel et al., 2022; Mohajerani et al., 2019; Zhang et al., 2019). One approach is to segment the images into different areas and extract the calving front as the border between segmentation areas (Hartmann et al., 2021; Zhang et al., 2019; Baumhoer et al., 2019; Periyasamy et al., 2022; Loebel et al., 2022). Another approach directly trains a model on the position of the calving front (Davari et al., 2022). This task suffers from severe class imbalance due to the thin calving front. They approach this problem by creating a distance map from every pixel to the front line. The network is trained on the distance map instead of the thin front line. The actual front-line prediction is then extracted during post-processing.

Other works use the segmentation network DeepLabv3 (Chen et al., 2018) to detect calving fronts. The main advantage of DeepLabv3 over U-Net is the atrous spatial pyramid pooling, which makes the network adaptable to different image resolutions (Zhang et al., 2021; Cheng et al., 2021).

Calving Front Machine (CALFIN) proposed by Cheng et al. (2021) segments optical and SAR images into the ocean-land zones and extract the calving front during post-processing with a topography map of the area. They apply MTL with a late-branching architecture. They use two labels: a binary ocean mask and a binary calving front mask. They achieve state-of-the-art predictions with an $86\,\mathrm{m}$ deviation from the measured calving front. A detailed comparison to aforementioned U-Net-based methods by Mohajerani et al. (2019) and Baumhoer et al. (2019) revealed the generalization ability of CALFIN on other glacier datasets. The images had to be down-sampled for CALFIN. Therefore, the distance in meters doubles but comparing the pixel distance errors reveals a similar performance. With $29\,\mathrm{M}$ parameters, CALFIN is still a large network that needs a large amount of training data. Chen et al. (2019) propose a similar approach for medical image segmentation and show that the individual tasks benefit from MTL. Several other approaches have advanced the U-Net architecture for MTL for medical applications (Abolvardi et al., 2020; Kholiavchenko et al., 2020; Li et al., 2019; Amyar et al., 2020). The work of Heidler et al. (2021) uses MTL for the segmentation of a binary ocean-land mask and edge detection of the Antarctic coastline, where the calving front is just part of the coastline. They add task-specific heads for the two tasks to the U-Net. They achieve results with a deviation from the reference of $345\,\mathrm{m}$ compared to $483\,\mathrm{m}$ with the vanilla U-Net. To avoid distortions of the metric from areas far from the coast, the metric is calculated within $2\,\mathrm{km}$ of the true coastline. A further development of the HED-UNet for Antarctic ice shelf front detection is proposed by Baumhoer et al. (2023). Their post-processing includes an elevation threshold of $110\,\mathrm{m}$ to remove erroneous classifications in the high-altitude dry snow zones.

## 3 Dataset

The dataset used in this work is provided by Gourmelon et al. (2022). It contains 681 Synthetic Aperture Radar (SAR) images of seven marine-terminating glaciers taken by six different satellites. Two glaciers are located in the northern hemisphere, namely Columbia (COL) Glacier in Alaska and Jakobshavn Isbrae (JAK) (Sermeq Kujalleq) in Greenland. The five glaciers in the

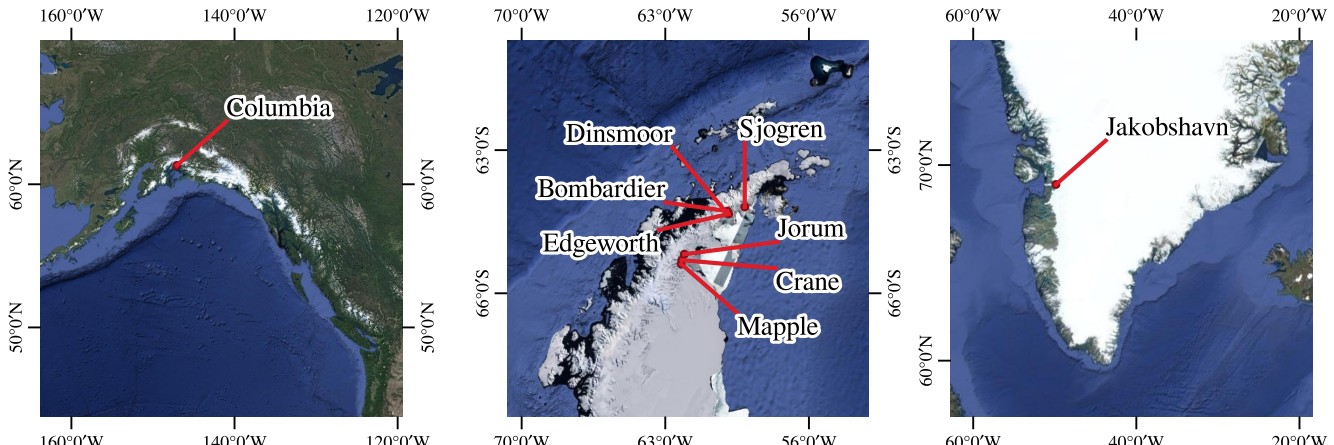

**Figure 2.** Location of the seven benchmark glaciers: Columbia Glacier in Alaska; Crane; Mapple; Jorum; Dinsmoore, Bombardier, Edgeworth (DBE); Sjogren Glacier in the Antarctic Peninsula; and Jakobshavn Isbrae Glacier in Greenland. The background images are taken from the Google Maps Satellite imagery layer © Google 2019.

| | Alaska | Antarctic Peninsula | | | | | Greenland |
|---|---|---|---|---|---|---|---|
| | Columbia | Mapple | Crane | Jorum | DBE | Sjögren-Inlet | Jakobshavn Isbrae |
| Split | - test: 122 - | - train: 559 - | | | | | |
| Images # | 65 | 57 | 69 | 77 | 133 | 121 | 159 |
| Area [km] | 32 x 15 | 8 x 8 | 19 x 25 | 20 x 13 | 22 x 20 | 23 x 19 | 16 x 19 |

**Table 1.** Properties of the dataset, including a list of captured glaciers, train-test split, number of images per glacier, and covered area.

southern hemisphere are all located on the Antarctic Peninsula (AP) (see Fig. 2). The Crane, Mapple, and Jorum glaciers are closest to the south pole, followed by Dinsmoore, Bombardier, Edgeworth (DBE). They are located so close together that they

are treated as one glacier site. The last glacier is the Sjögren-Inlet (SI) Glacier.

Table 1 lists the seven glacier sites with the number of images and the area they show. The table also depicts the train-test split. The samples of the train set are used for the parameter optimization of the segmentation method, and the test set is exclusively used for evaluating the method. The test set contains the glaciers Mapple and Columbia. Columbia is the only mountain glacier in the dataset. The other glaciers are outlet glaciers of polar ice sheets. Therefore, Columbia can be seen as a benchmark glacier

for the generalization capability of the segmentation method. Columbia and Mapple also differ in the shape of the glacier. The images of Columbia show multiple calving fronts in one image, and the flow of the glacier arms goes in different directions. On the other hand, Mapple is a less complex glacier, moving in only one direction with one clearly defined calving front between the lateral fjord side walls.

The dataset contains two labels and a bounding box of the glacier for each SAR image. One shows a mask of the different

zones of the glacier (ocean, glacier, no information available, rock). The other label contains a one-pixel wide line representing

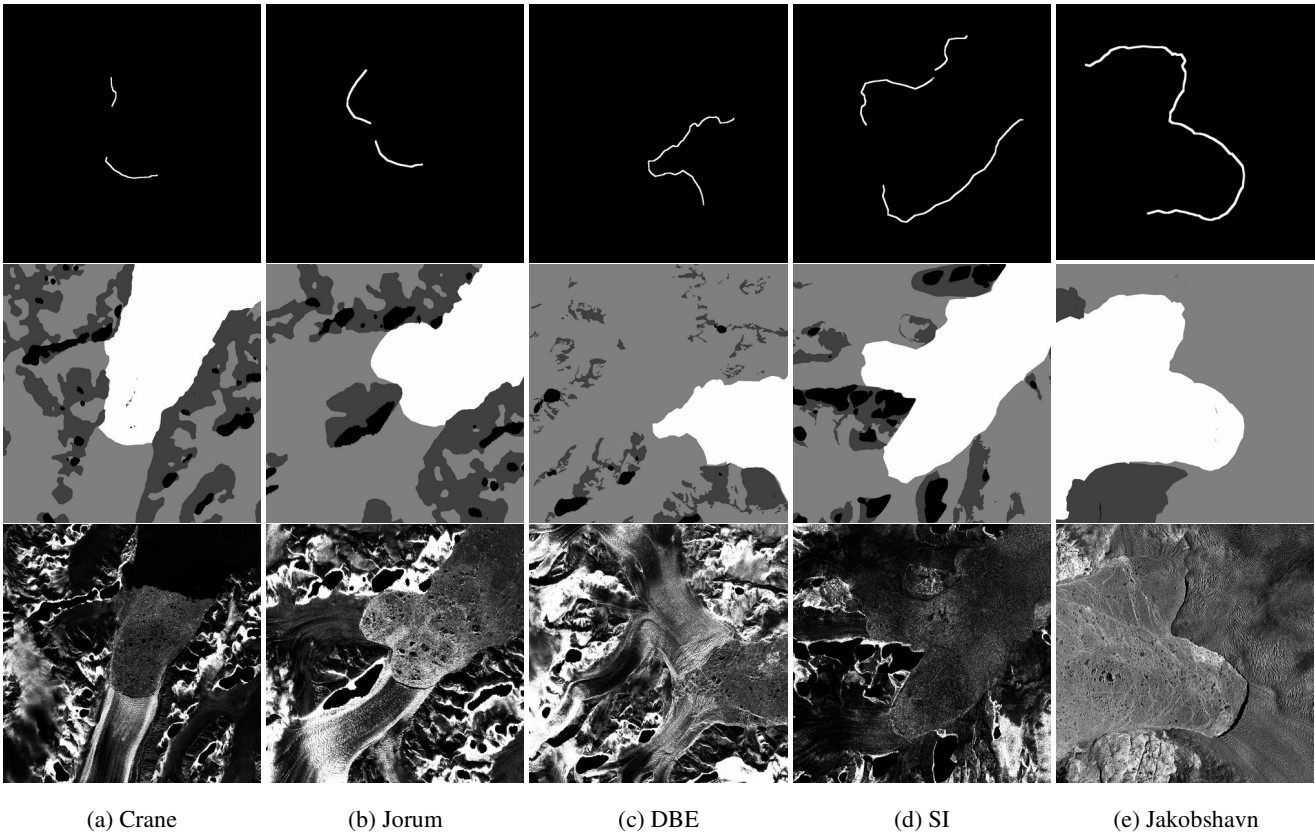

|   (a) Crane   |   (b) Jorum   |   (c) DBE   |   (d) SI   |   (e) Jakobshavn   |

**Figure 3.** Sample images of every glacier in the training set and their corresponding labels. The first row shows the front label with a black background and a one-pixel wide white line representing the calving front. The second row contains the zone labels with four classes: ocean (white), glacier (light grey), rock (dark grey), and no information available (black). Black dots in the ocean zone of Crane are areas with no data available due to a small artefact at the former calving front of Crane Glacier in the DEM (Cook et al., 2012) used for the orthorectification of the SAR data. SAR imagery is provided by DLR, ESA, and ASF.

the calving front. The bounding box is not utilized for our method. A sample of each glacier in the training set with its corresponding labels is shown in Fig. 3. Predicting the zone mask can be seen as a classic segmentation problem. The calving front can also be extracted from the zone label by taking the border between ocean and ice areas in the corresponding bounding box. The direct delineation of the calving front is a more difficult task due to the high-class imbalance. Fewer than 1 % of the pixels are labelled as front pixels. Additionally, the structure of the class region is not a convex hull but a thin line.

Corresponding to the change of glacier area, the position of the calving front changes. In Fig. 4, the retreat of Columbia is visualized by plotting the calving front position of every sample in a different colour. The bright lines represent past calving fronts, and the dark red lines the more recent positions. The constant retreat of COL is visible through the tiered lines. The

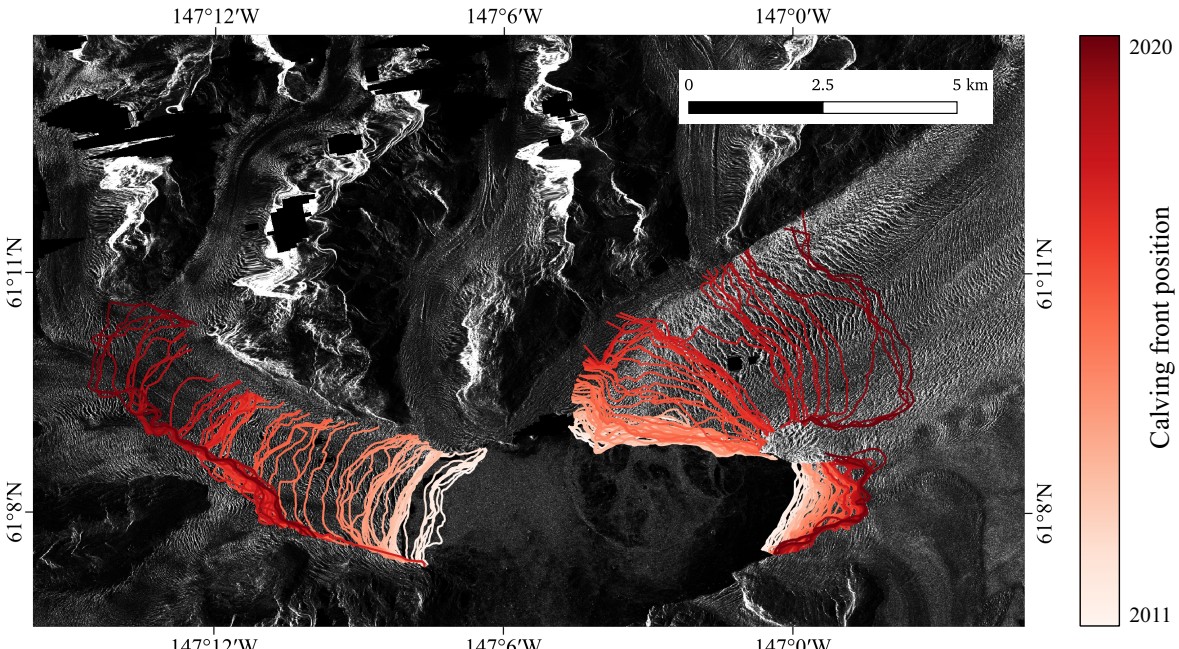

**Figure 4.** Evolution of the calving front position of Columbia Glacier (Alaska). The red lines are all calving front labels available in the dataset. Background: SAR intensity image acquired by TDX on date 13 November 2011. Projection CRS EPSG:4326 - WGS 84 / UTM zone 6N. SAR imagery is provided by DLR, ESA, and ASF.

visualization of the glaciers in the southern hemisphere is included in Appendix B. The change in the class distribution of the zone labels over time is shown in Appendix. C.

Every glacier is captured by multiple satellites for a higher temporal resolution and extended observation periods. Meaning that recordings of one glacier are captured by different SAR systems with different image resolutions. The resolution refers to the ground range resolution. Environmental Satellite (ENVISAT), European Remote-Sensing Satellite-2 (ERS), Sentinel-1 (S1) has a resolution of 20 m, Phased Array type L-band SAR (PALSAR) has a 17 m resolution, and TDX has a 7 m per pixel resolution. Unfortunately, the dataset does not provide information about the polarization. In Fig. 5, a timeline of the images of each glacier visualizes the observation time and frequency of the images. The first two rows show the glaciers of the test set.

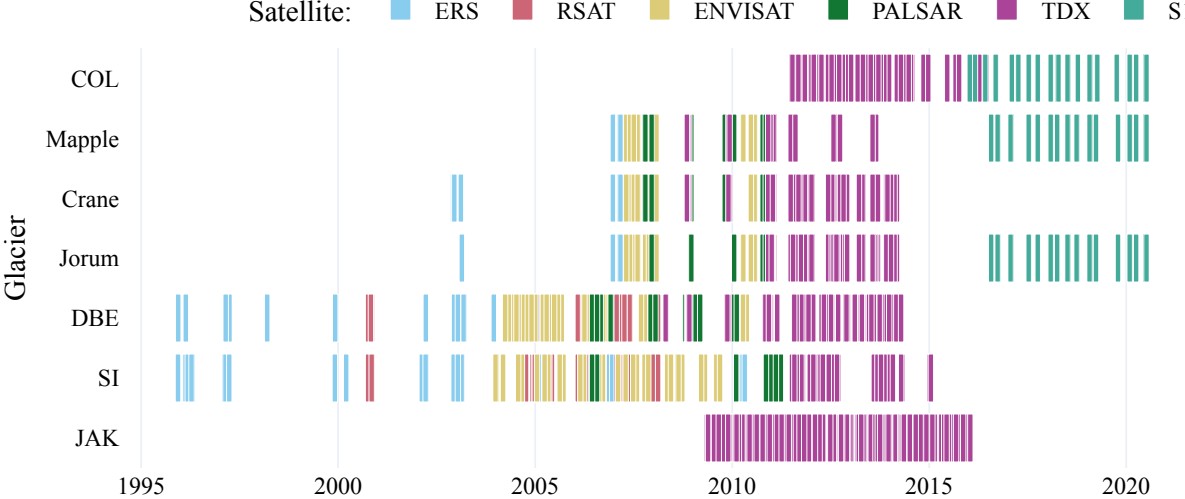

**Figure 5.** Distribution of the dataset's images over time. The samples are grouped by the seven glaciers and coloured according to the capturing satellite. We abbreviated the following glaciers: Columbia (COL), Dinsmoore, Bombardier, Edgeworth (DBE), Sjögren-Inlet (SI), Jakobshavn Isbrae (JAK).

## 4 Method

In this section, we explain our method, which includes the utilisation of the nnU-Net as a framework that simplifies the training of a U-Net. We document our six experimental setups that aim to evaluate the impact of MTL on the training of the U-Net.

### 4.1 Background

In the field of deep learning, a lot of time and effort is put into the hyperparameter search. Isensee et al. (2021) proposes the nnU-Net that automates the manual tuning of hyperparameters. The nnU-Net is a framework around the U-Net architecture. It provides good default values for hyperparameters, rules to adapt hyperparameters to the dataset, and many established deep-learning techniques for training and inference. Most of the rule-based hyperparameters are only relevant for three-dimensional (3D) data in the medical domain, like Computer Tomography (CT) and Magnetic Resonance Imaging (MRI) and are irrelevant to our experiments. For all image modalities except CT, z-score normalization is applied. Each image is normalized independently by first subtracting its mean and then dividing by its standard deviation. Another dataset parameter is the median shape of the samples. The shape and the available Graphics Processing Unit (GPU) memory determine the patch and batch size and, therefore, the network topology. The patch size is initialized with the median shape and iteratively reduced while adapting the network topology accordingly until the network can be trained with a batch size of at least two given GPU memory constraints.

In addition to the rule-based parameters, there are fixed parameters. These parameters are based on the authors' experience and generalize well across various tasks. The nnU-Net uses a poly learning rate scheduler, a combination of dice coefficient and cross-entropy as the loss function, and Stochastic gradient descent (SGD) with a Nesterov momentum as optimizer. The dice

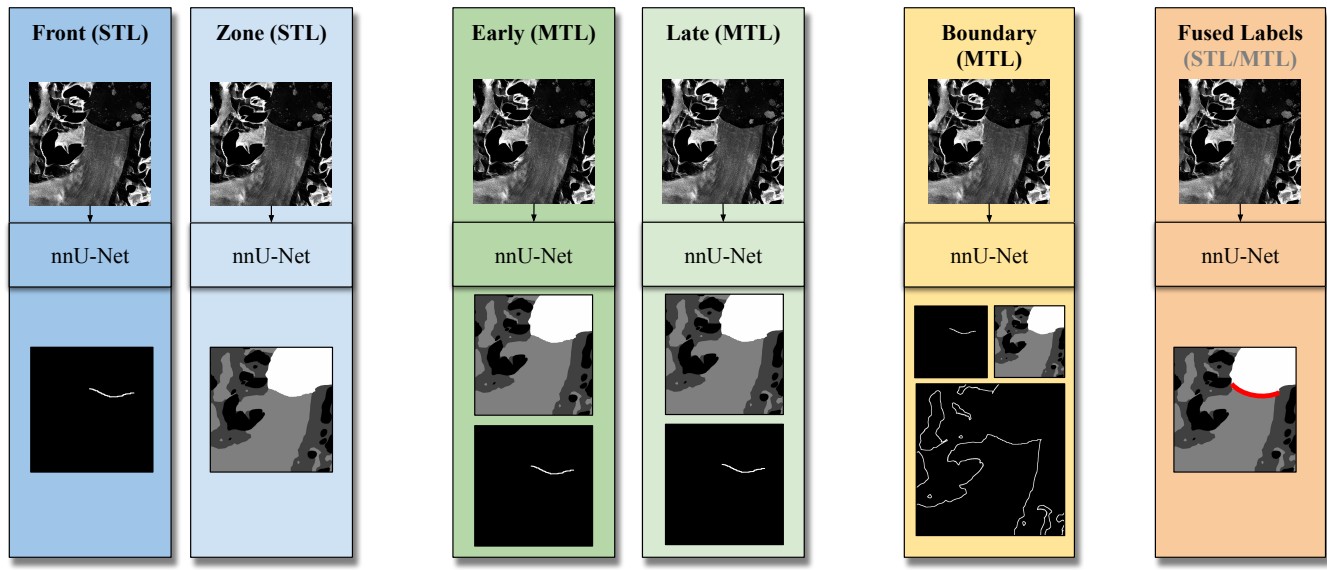

**Figure 6.** Illustration of the six experiments: Two Single-Task-Learning (STL) experiments (blue), two MTL experiments with different architectures (green), one MTL experiment with an additional task of delineating the whole glacier boundary (yellow), and one experiment where the two labels are fused into one segmentation task with an additional class in the zone label (orange).

coefficient also helps with the problem of class imbalance. The coefficient of each class is weighted by the number of pixels in the label that relate to the class. Additionally, deepsupervision is used to avoid vanishing gradients in neural networks with many layers. Independent of the dataset size, one epoch is defined as 250 mini-batches with foreground oversampling. The foreground oversampling is especially helpful for our application because the class imbalance between the calving front and background is high. It ensures that (at least) one-third of the patches for training are guaranteed to contain a foreground class. In our case, every batch contains two patches, from which one is forced to contain calving front pixels.

This framework achieves robust results for various medical image segmentation tasks. Overall, nnU-Net sets a new state of the art in 33 of 53 segmentation tasks of the Kidney and Kidney Tumor Segmentation Challenge challenge (Heller et al., 2021) and otherwise shows performances on par with or close to the top leaderboard entries. But the performance of the nnU-Net on segmenting glacier SAR images is yet to be tested.

### 4.2 Multi-Task Learning with nnU-Net

The first research goal is to apply the nnU-Net out-of-the-box on the glacier front detection and on the glacier zone segmentation, respectively. Training the nnU-Net directly on the front labels is the most straightforward approach for calving front detection. The nnU-Net is intended to be used in the STL manner. In Fig. 6, these two baseline experiments are represented by the two blue columns on the left. The label of the calving front is dilated to the width of five pixels. Our preliminary experiments have

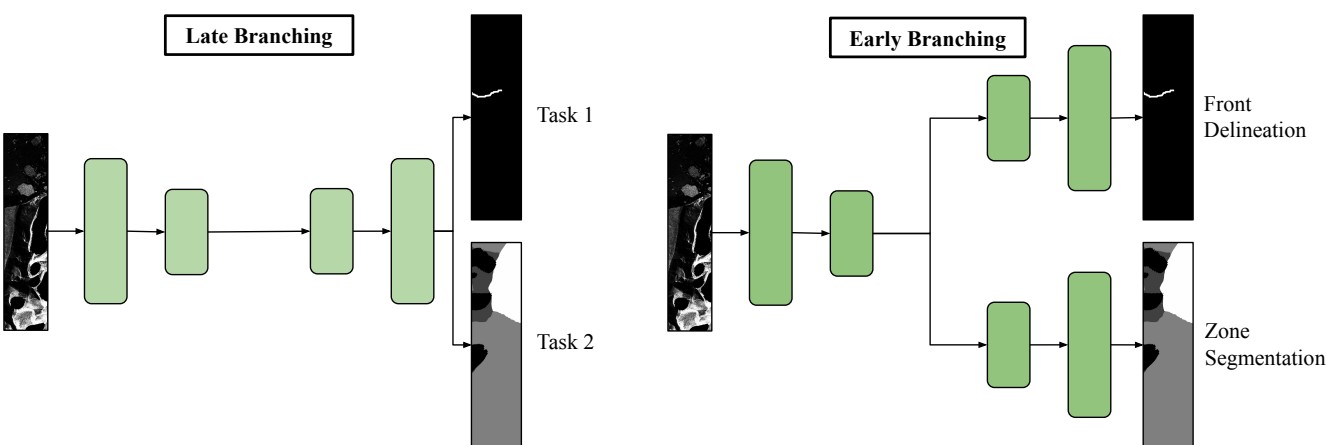

**Figure 7.** Illustration of early- and late-branching U-Net architectures for MTL. Both architectures perform joint feature extraction. The early-branching architecture applies separate reconstruction with two decoders, one for each task. The late-branching network applies joint reconstruction with a common decoder.

shown that dilation makes the predictions more robust. For the training with zone labels, the post-processing includes extracting
the boundary between the ocean and the glacier.

In the following experiments, the segmentation problem changes to a multi-task problem, where both labels are used to train one model. The next two experiments concern network architecture. They are represented by the two green columns in Fig. 6. The early-branching architecture uses one decoder for every label. Thus, the number of parameters increases by about $50\,\%$. In contrast, the late-branching architecture requires only a small change to the vanilla U-Net. An additional channel of the last
layer is used to predict the second label. For this architecture, only the weights of one kernel are trained additionally to the set of parameters needed for one task. The change in the total number of parameters that need to be trained is negligible.

Because architecture changes with multi-task learning were not foreseen in nnU-Net framework, we had to make changes to the framework. During the experiment's planning, i. e., pre-processing phase, we fixed the network architecture's estimated size to the size of the early-branching network so that late- and early-branching networks return the same value for the network size.
Otherwise, early and late branching networks would be trained on different patch sizes. Thus, performance differences during the evaluation might arise from the different patch sizes and not the differing architectures. During training, the error of all labels is calculated and summed up with equal weighting. For the inference, we adapted only the number of channels in the case of MTL. After the test samples are divided into patches and fed through the network and the patch predictions are combined into the prediction of the whole image. The predictions of the zones are post-processed to get an additional result for the position of
the calving front. All glacier pixels with a neighbouring pixel classified as the ocean are classified as glacier front to extract the glacier front from the zone predictions. We note that the prediction for respectively other task is omitted in the inference.

The last two experiments concern label changes. In Fig. 6, they are coloured yellow and orange. The fifth experiment of this work, cf. Fig. 6 (yellow), extracts the boundaries between the glacier zone and all other zones as a third segmentation task

for the late-branching U-Net. The label of the glacier boundaries was extracted from the zone label. All glacier pixels with a neighbouring rock or shadow pixel are assigned as glacier boundaries. The hypothesis is that providing more information about the same sample benefits the performance of the U-Net on the individual tasks. The third segmentation task is not considered in the final evaluation. The last experiment fuses the zone and front labels by creating a fourth class in the zone label associated with the glacier front, cf. Fig. 6 (orange). As the front line has a width of five pixels ($35 - 100\,\mathrm{m}$ depending on the image resolution), the other zone classes are merely impaired. A post-processing step is added to the predictions. The front pixels are isolated to generate a front prediction for comparing the results with the other experiments. To generate a prediction of the zones, pixels classified as front are assigned to the ocean, and the glacier zone is dilated once with a 7x7 kernel.

## 4.3 Experimental Setup

The nnU-Net was trained on a NVIDIA RTX 3080 with 12GB memory; the adapted network architecture has nine encoder blocks and eight decoder blocks. Each block consists of two convolutional layers: An instance normalization and a Rectified Linear Unit (ReLU). The kernels of all convolutional layers have a size of 3x3. During training, one batch contains two images. The patch size of the experiments that include only one label is 1280x1024. The experiments that have more than one label have a patch size of 1024x896 because of the GPU memory limit. There are also fixed parameters that are independent of the dataset. This includes the SGD optimizer with an initial learning rate of 0.01, a Nesterov momentum of 0.99, and a weight decay of 3e-5. Training of one epoch took between 100 s and 160 s. The maximum number of epochs of 500 is reached in every training (due to limited resources, we reduced the original maximum number of epochs from 1000 to 500). The nnU-Net defines one epoch using a fixed number of iterations (250). In each iteration, the batch is sampled depending on the class distribution of the sample to counteract the class imbalance.

## 5 Evaluation

In this section, we will examine our evaluation metrics, compare the results of the six proposed experiments and evaluate the results of the fused label experiment. We used a five-fold cross-validation for the evaluation to eliminate the weight initialisation bias and the data-split bias into training and validation sets. The metric scores of the individual models are averaged to get a robust measure independent of weight initialization and split.

### 5.1 Evaluation Metrics

As our main measure, we use the Mean Distance Error (MDE) proposed by Gourmelon et al. (2022). It measures the distance of the predicted front $\mathcal{P}$ to the ground truth calving front $\mathcal{Q}$ for all images in the test set $\mathcal{I}$. For every pixel in the label front $\mathcal{Q}$, the distance to the closest pixel in the predicted front $\mathcal{P}$ is determined. Additionally, to make the metric symmetric, the distance to the closest pixel in the label front $\mathcal{Q}$ is determined for every pixel in the predicted front $\mathcal{P}$. These distances are averaged and taken as the mean distance between the two lines (see Fig. 8 and Eq. 1). We note that the front MDE is also calculated for the

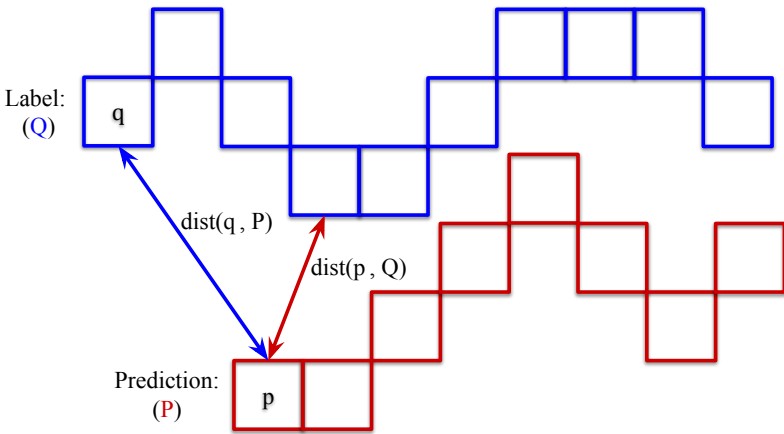

**Figure 8.** Visualization of the Mean Distance Error (MDE) calculation between front label $Q$ and front prediction $P$. The $dist(q, P)$ represents the distance between $q$ and the pixel in $P$ that is closest to $q$.

zone segmentation. We extract the front from the zone segmentation by assigning all glacier pixels with neighbouring ocean pixels as the glacier front.

$$\text{MDE}(\mathcal{I}) = \frac{1}{\sum_{(\mathcal{P},\mathcal{Q})\in\mathcal{I}}(|\mathcal{P}|+|\mathcal{Q}|)} \sum_{(\mathcal{P},\mathcal{Q})\in\mathcal{I}} \left( \sum_{p\in\mathcal{P}} \min_{q\in\mathcal{Q}} ||p-q||_2 + \sum_{q\in\mathcal{Q}} \min_{p\in\mathcal{P}} ||p-q||_2 \right) \tag{1}$$

Additionally, we give classical segmentation metrics to evaluate the zone prediction. They include the Intersection over Union (IoU), which is defined as $\text{IoU} = \frac{T_P}{T_P+F_P+F_N}$, i.e., the True Positive ($T_p$) pixels over the sum of $T_p$, False Positive ($F_p$), and False Negative ($F_N$) pixels. Additionally, the F1-score is computed ($F1 = 2 \cdot \frac{pr \cdot re}{pr+re}$), which is a combination of Recall ($re = \frac{T_p}{T_p+F_N}$) and Precision ($pr = \frac{T_P}{T_P+F_P}$). Because this segmentation task is not binary, but every sample has four classes, the metrics are calculated for each class individually and then averaged with equal weighting.

### 5.2 Results and Discussion

In this section, we evaluate our experiments. In section 5.2.1, we compare the performance of the six experiments, which we described in section 4.2. In section 5.2.2, we analyse the prediction of the fused label experiment and investigate the influence of season, glacier site and satellite on the calving front prediction performance.

#### 5.2.1 Comparison of the five experiments

In Fig. 9, the MDE of every experiment is compared. The STL approach that is trained on the front labels has an MDE of $1103 \pm 72\,\text{m}$ and the STL approach that is trained on the zone labels has an MDE of $1184 \pm 225\,\text{m}$. A difference between the STL experiments arises in the performance variance, where the model trained on the zone labels is larger. The number of

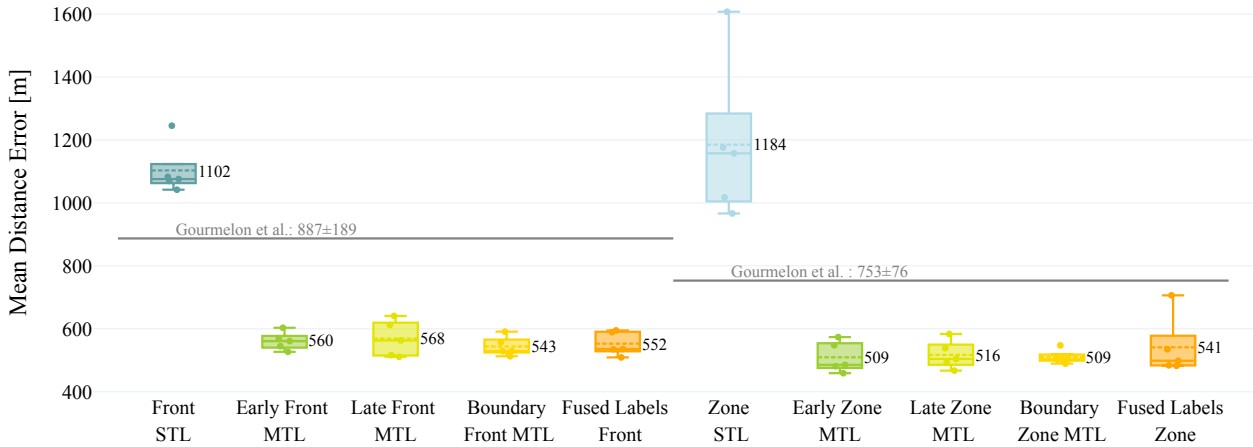

**Figure 9.** Box plot of the MDE of the six experiments. Two columns are coloured equally for each experiment except for the STL experiments. The left half represents the front delineation based on the front segmentation (Task 1), and the right half represents the front delineation based on the zone segmentation (Task 2). The baseline of Gourmelon et al. (2022) is displayed as a grey line in each half.

samples that are incorrectly predicted to have no front pixel is similar, with an average of $27.2 \pm 6.0$ for training on the front and $24.8 \pm 12.4$ out of 122 for training with zone labels.

The lowest number of samples with false non-front-detections achieves the model that is trained on the fused labels, $3.2 \pm 2.0$ for the front label out of 122 samples. The baseline of Gourmelon et al. is $1 \pm 1$ based on the zone segmentation and $7 \pm 3$ from the front prediction. All values can be seen in Table A2. All MTL models have a significantly lower MDE than STL models with a significance level of $\alpha = 0.01$ using a student's t-test. The Table with the T-values of all experiment pairs is given in Table A1. The model that is additionally trained on the glacier boundary has the smallest MDE for both tasks. The MDE baseline of Gourmelon et al. is $887 \pm 189$ m for the front prediction and $753 \pm 76$ m for the zone. Overall the MDE is similar for all MTL approaches. The student's t-test shows that the differences between fused label and all other MTL approaches are insignificant ($\alpha = 0.33$).

The metrics for the zone segmentation, shown in Table A3, show a similar trend. An improvement of all MTL over the STL approach and minor changes between the MTL approaches. The STL of the nnU-Net on zone label achieves $62.4 \pm 3.5$ IoU and $71.7 \pm 3.2$ F1-score. The highest F1-score achieves the model that is trained with the additional boundary task with $81.7 \pm 0.5$. The model's IoU is $72.6 \pm 0.4$. The baseline of Gourmelon et al. is $80.1 \pm 0.5$ F1-score and $69.7 \pm 0.6$ IoU.

The fused label approach is the most feasible method discussed as follows. This approach performs on par with the other experiments. Even though the MDE of the boundary experiment is slightly lower than for the fused label approach, the student t-test showed that the difference is not significant. Moreover, the fused label approach requires fewer parameters and no changes

to the nnU-Net framework, making its use truly out-of-the-box. Additionally, this approach has a relatively small number ($5 \in 122$) of predictions with falsely non-detected fronts.

### 5.2.2 Analysis of the fused label experiment

For the final evaluation, the zone segmentations of the five models that are trained during the five-fold cross-validation on the fused labels are combined into a more robust ensemble prediction. The models differ by their weight initialization and train-validation split. But are all trained on the fused labels. The ensemble prediction is created by taking the mean of the probabilities for every class each model gives. We don't use the calving front pixels directly from the prediction but apply the post-processing of the zone label to gain a slightly better MDE. The post-processing labels the edge of the glacier zone as a

calving front that has neighbouring ocean pixels. The performance of the ensemble prediction results in an MDE of $515 \pm 39\,\mathrm{m}$ and $5 \in 122$ predictions with no fronts, which is similar to the average performance of the single models. We did not use the ensemble prediction for the comparison between the experiments in section 5.2.1 to show the variance that is introduced by a different train-validation split and different weight initialization.

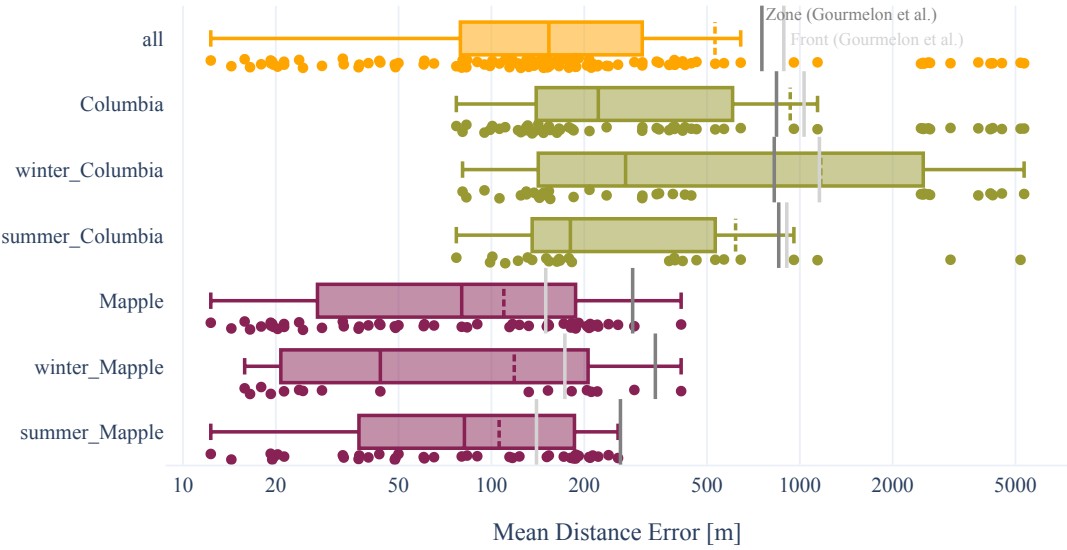

**Figure 10.** MDE on the test set grouped by glaciers and subdivided into seasons. The orange box plot in the first row shows all MDE of the test set. The olive green plot below shows the errors from the COL images, subdivided into seasons in the third and fourth rows. The last three rows show errors from the Mapple images. The dark grey line shows the baseline of the zone prediction, and the light grey line shows the baseline of the front prediction from Gourmelon et al. (2022). The ensemble model was trained with the fused zone and front labels. The y-axis has a logarithmic scale. The median is the middle line in the rectangle, and the dashed line represents the mean. The x-axis has a logarithmic scale. Otherwise, the outliers would dominate the plot. The rectangle reaches from the first quartile to the third quartile.

   The distribution of the MDE on the test set predictions is plotted in Fig. 10. In the first row, all errors of the prediction of the

117 test samples are drawn as dots. The test set contains two glaciers: Mapple and COL. The rows below show the distribution

of the MDE on the two test glaciers separated into summer and winter seasons. The main difference in MDE is between glaciers, similar to the baseline results of Gourmelon et al. (2022) with MDE of $287 \pm 48\,\text{m}$ for Mapple and $840 \pm 48\,\text{m}$ for COL. The MDE of our methods is on average $109 \pm 90\,\text{m}$ for Mapple, while the MDE of COL is $930 \pm 1420\,\text{m}$. The difference of the MDE is caused by a group of predictions with an error $> 1000\,\text{m}$. The median value is $222\,\text{m}$ for COL and $80\,\text{m}$ for Mapple. The reasons for the large glacier differences might be related to the different shapes. Mapple has a simple calving front, a single line constraint by a straight valley. Columbia has multiple calving fronts, a stream coming from the left side of the image, one from the top, and another from the left (see Fig. 13).

There is also a seasonal difference in the MDE. The MDE of the front prediction during the summer of Mapple and COL combined have lower MDE $307 \pm 730\,\text{m}$ than the samples captured during the winter months ($818 \pm 1389\,\text{m}$). However, the medians are closer together with $150\,\text{m}$ in the summer months and $181\,\text{m}$ in the winter months. The MDE baseline (Gourmelon et al., 2022) in summer is $732 \pm 93\,\text{m}$ and $776 \pm 65\,\text{m}$ in winter. Although, most outliers are from winter seasons in COL. In winter, snow coverage of the glacier is more likely. The SAR signal can penetrate through several meters of snow cover, depending on the SAR frequency and the water content of the snow cover. However, most of the studied glaciers, in particular Columbia Glacier and the glaciers on the Antarctic Peninsula, are located in temperate marine environments making wet snow conditions also likely during winters. Thus snow can cover useful artefacts like crevasses and rock structures that can be useful for the pattern recognition of the nnU-Net. However, more important is the fact that the ocean next to the glaciers is covered more often by ice mélange during winters, which reduces the contrast between ocean and glacier areas. Even for experienced human mappers, it can be a challenging task to distinguish between the glacier tongue and the ice mélange. Thus, we hypothesize that the network has similar issues leading to reduced performance during winter. The difference between seasonal MDE also depends on the sensor. The MDE of the low-resolution sensors (S1, ENVISAT) is lower in the summer month. The MDE of high-resolution sensor (TDX) is lower in winter months but the difference is much smaller. A table of the MDE grouped by satellite and season is in Appendix A4.

An overview of the impact of the sensor resolution is given in Fig. 11. The images with a resolution of $7\,\text{m}$ per pixel have a mean MDE of $180\,\text{m}$ resolution have a mean MDE of $135\,\text{m}$, but this class contains only images of Mapple glacier taken by PALSAR. The images with $20\,\text{m}$ per pixel have a mean MDE of $1214\,\text{m}$. The distribution has a cluster of MDEs that is larger than $1000\,\text{m}$. These large errors only stem from images of COL (see Fig. 12 row Columbia_S1).

In Fig. 12, the MDE is grouped by satellites. The predictions for the samples captured by ERS, ENVISAT, and PALSAR have a similar average error between $150\,\text{m}$ and $300\,\text{m}$. However, the samples from ERS, ENVISAT, and PALSAR only represent Mapple Glacier. TDX captures both test sites an has a MDE of $180 \pm 179\,\text{m}$ The samples captured by S1 have a higher MDE with $1664 \pm 1807\,\text{m}$. The outliers are all from samples captured by S1 of Columbia. The neural network can not generalise from the training set to this set of samples. The 14 outliers have a front-line delineation error of $> 1000\,\text{m}$. These are predictions from images of COL taken by S1. The low-resolution, e. g., cracks in the ice, are not visible. Fig. 13 (c) shows that the model falsely predicts ocean pixels as the front and does not predict the left calving front at all.The MDE for this sample is $4221 \pm 5026\,\text{m}$. We suspect that the complex calving front of COL requires a high resolution for accurate detection or more training data of S1. The group of outliers heavily increases the overall MDE but can be separated clearly by specifying satellite and glacier. It shows

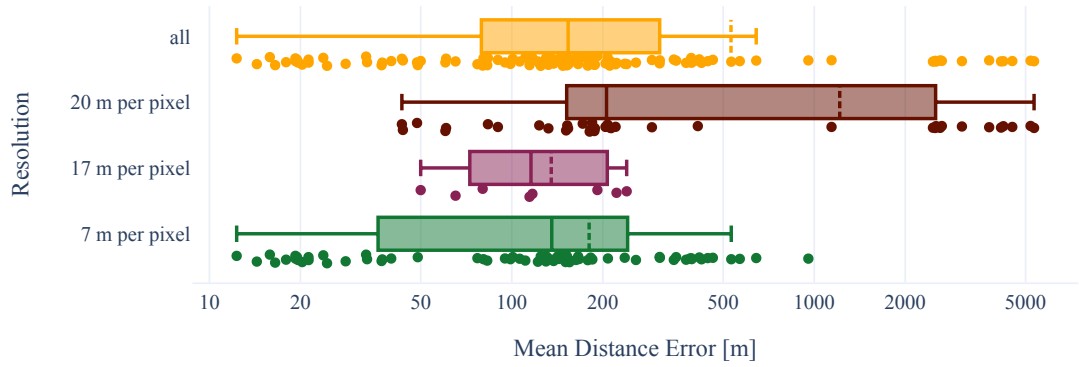

**Figure 11.** MDE on the test set grouped by resolution of the SAR image. The orange box plot in the first row shows all MDE of the test set. ENVISAT, ERS and S1 have a resolution of 20 m per pixel. MDE of this resolution are represented by the dark red box plot. PALSAR has a 17 m per pixel resolution. MDE of images of this resolution are represented in the magenta box plot. TDX has a 7 m per pixel resolution. MDE of images of this resolution are represented in the green box plot. The ensemble model was trained with the fused zone and front labels. The dark grey line shows the baseline of the zone prediction, and the light grey line shows the baseline of the front prediction from (Gourmelon et al., 2022). The y-axis has a logarithmic scale. The median is the middle line in the rectangle, and the dashed line represents the mean. The x-axis has a logarithmic scale. Otherwise, the outliers would dominate the plot. The rectangle reaches from the first quartile to the third quartile.

the generalisation limits of our method. For a visual inspection, we provide the predictions of the test set as single images and as an animation on Zenodo (https://zenodo.org/record/8379954).

The MDE is negatively influenced by calving front predictions extending over the labelled calving front on the coastline, which can be seen in Fig 13 (a) and (b). The MDE might give a wrong implication on the usability of these predictions. The prediction

of 13 (b) has a relatively high MDE (393 m) because it predicts the coastline as a calving front. Despite this, the prediction can still be used to monitor the change in the position of the calving front as well as the calculation of the total frontal ablation rate or surface area. The prediction can be corrected easily if the coastline is known. The misclassification of the coastline as a calving front holds true for nearly all images of COL taken by TDX. The northern part of COL is particularly difficult to distinguish between the coastline and calving front because the glacier retreated so far that it transitioned from a marine-terminating glacier

to a land-terminating glacier. Without prior knowledge, it is even difficult for humans to distinguish between the coastline and the calving front.

The model predictions of the calving front on images of Mapple have a high overlap with the label, even with a low resolution of ≥ 20 m (ERS) (see Fig. 14). There are some outliers for Mapple as well, but they are not as severe as the outliers at COL. The prediction of Mapple with a high resolution of 7 m per pixel is close to the ground truth (Fig. 14).

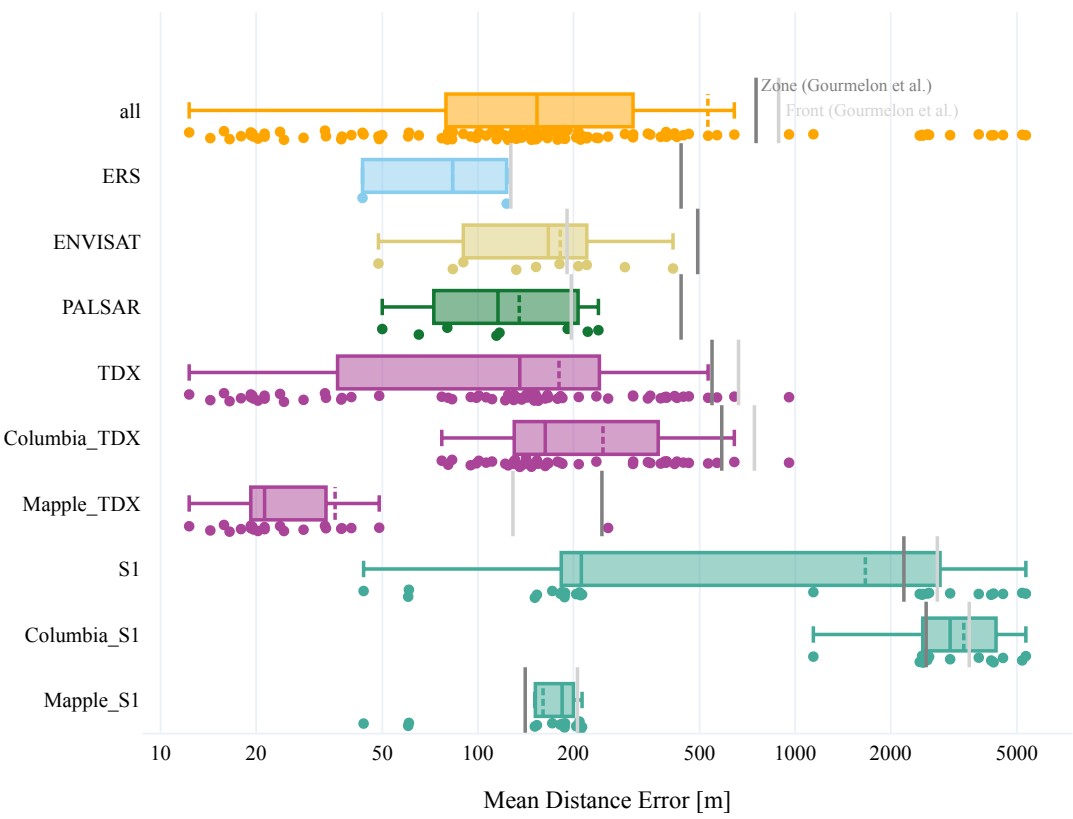

**Figure 12.** MDE on the test set grouped by satellites. The dark grey line shows the baseline of the zone prediction, and the light grey line shows the baseline of the front prediction from (Gourmelon et al., 2022). The images from ERS (light blue), ENVISAT (yellow), and PALSAR (green) only capture the Mapple Glacier. The MDE from TDX (pink) and S1 (turquoise) is subdivided into the two glaciers of the test set. The ensemble model was trained with the fusion of zone and front labels. The y-axis has a logarithmic scale. The median is the middle line in the rectangle, and the dashed line represents the mean. The x-axis has a logarithmic scale. Otherwise, the outliers would dominate the plot. The rectangle reaches from the first quartile to the third quartile.

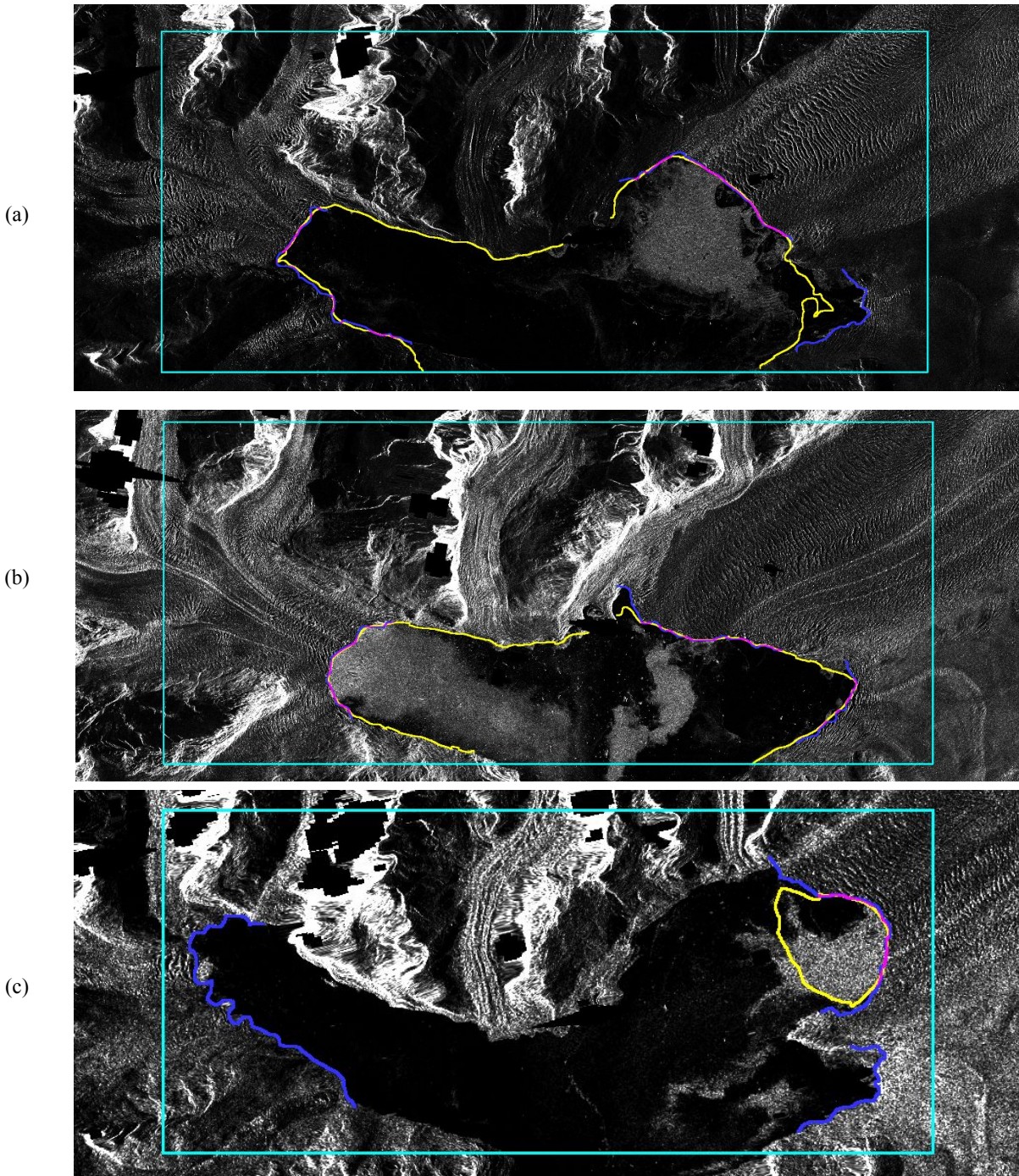

**Figure 13.** Calving front predictions of COL on (a) 11 February 2016, and (b) 22 June 2014 taken by TDX with 7m pixel resolution (after multi-looking with a factor of 3x3); (c) was taken by S1 on 10 September 2019 with 20 m pixel resolution; ground truth (blue), prediction (yellow), the overlap of ground truth and prediction (magenta). SAR imagery is provided by DLR, ESA, and ASF.

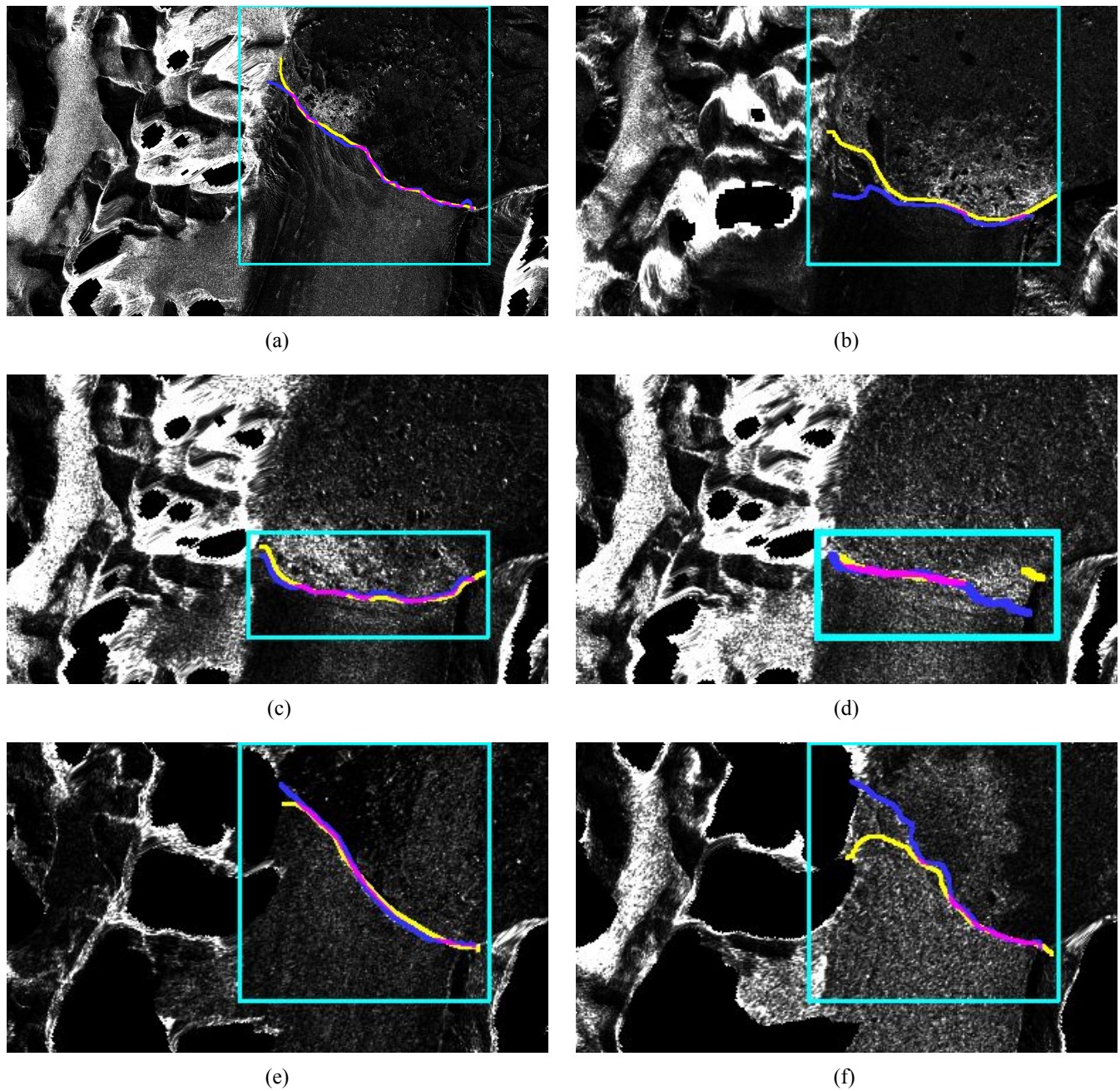

**Figure 14.** Calving front predictions of Mapple Glacier; ground truth (blue), prediction (yellow), the overlap of ground truth and prediction (magenta), bounding box (cyan). (a) was taken by TDX on 13 October 2008. (c) was taken by PALSAR on 29 November 2008. (c) was taken by S1 on 3 July 2016. (d) was taken by S1 on 2 March 2019. (e) was taken by ERS on 5 February 2007. (f) was taken by ENVISAT on 22 September 2007. SAR imagery is provided by DLR, ESA, and ASF.

# 6 Conclusion

This work explores the use of the nnU-Net by Isensee et al. (2021) to segment glacier calving fronts. The nnU-Net promises an out-of-the-box application of deep learning for segmentation tasks. We evaluate this claim on a dataset of glacier images provided by Gourmelon et al. (2022). The dataset contains two tasks. The calving front detection and the glacier zone segmentation. We try different modifications of Multi-Task-Learning with two different neural network architectures to tackle both tasks simultaneously. The results show that combining both tasks increases each task's performance. No significant difference between the two Multi-Task-Learning architectures exists. Adding more domain-specific tasks like glacier boundary delineation does not further improve the previous tasks. Due to the small area of the front line, the two labels can be fused into one label. The fusion of labels decreases the number of parameters used, shortens the training time, and reduces the deep learning expertise needed, as the nnU-Net can be used without modifications. With this approach, we achieve an average MDE of $541 \pm 84\,\mathrm{m}$. We provide the code and the pre-trained model for application on further SAR images of glacier fronts (see Code and data availability). The predictions need to be filtered manually since there can be outliers, as our results show. But it will provide an initial prediction, which eases the task of glacier front delineation.

To improve the average MDE, future work should focus on reducing the model performance on low-resolution images like S1 with $20\,\mathrm{m}$ per pixel resolution. This can be done by including more images taken by S1 in the training set or by implementing an oversampling strategy of the low-resolution images.

The framework nnU-Net is well suited for segmenting SAR images of glaciers and calving front delineation. The modification of the nnU-Net for MTL improves the results compared to STL experiments, where only one label is used. However, it is not necessary for glacier segmentation and calving front delineation because both labels can be fused without losing a significant amount of information. Our findings highlight the suitability of the nnU-net for glacier front segmentation on multi-mission SAR remote sensing data, which will facilitate an efficient, extended spatiotemporal mapping of tidewater glacier terminus changes. Our findings also promote the out-of-the-box application of the nnU-Net for other segmentation tasks based on satellite imagery, because we didn't need to modify it for the calving front detection.

## Appendix A: Evaluation results of all experiments

This section provides more detailed values of the evaluation. Table A1 contains the T-values for stating significant and insignificant differences between models. Table A2 contains the zone MDE and the front MDE of the six experiments. The MDE is averaged over the five model results. Additionally, the Baseline from (Gourmelon et al., 2022) is provided in the first two rows. The values correspond to Fig 9. We provide Fig. table A5 to compare the two methods (nnU-Net and (Gourmelon et al., 2022)). Table A4 contains the MDE of the fuse label training grouped by satellite and season.

|  | Early MTL | Late MTL | Boundary MTL | Fused Labels |
|---|---|---|---|---|
| STL | 15.78 | 13.46 | 16.14 | 14.29 |
| Early MTL | 0 | 0.31 | 1.02 | 0.43 |
| Late MTL |  | 0 | 0.95 | 0.58 |
| Bound. MTL |  |  | 0 | 0.46 |

(a) Front position extracted from front prediction.

|  | Early MTL | Late MTL | Boundary Labels | Fused |
|---|---|---|---|---|
| STL | 6.57 | 6.51 | 6.66 | 5.96 |
| Early MTL | 0 | 0.29 | 0.03 | 0.75 |
| Late MTL |  | 0 | 0.35 | 0.58 |
| Bound. MTL |  |  | 0 | 0.81 |

(b) Front position extracted from zone prediction.

**Table A1.** T-value for the significance of difference. T-values that surpass the threshold of 3.36 means that the probability for the difference to be random is below 1 %. T-values below 1 mean that the difference is by chance, with a probability of 35 %, meaning that the difference is insignificant.

| Model | Experiment | Modality | MDE Front ↓ [m] | $\emptyset$ Front | MDE Zone ↓ [m] | $\emptyset$ Zone |
|---|---|---|---|---|---|---|
| Gourmelon et al. | Front | STL | $887 \pm 189$ | $7 \pm 3$ | - | - |
|  | Zone | STL | - | - | $753 \pm 76$ | $1 \pm 1$ |
| nnU-Net | Front | STL | $1102 \pm 72$ | $27.2 \pm 6.0$ | - | - |
|  | Zone | STL | - | - | $1184 \pm 255$ | $24.8 \pm 12.4$ |
|  | Early | MTL | $560 \pm 25$ | $8.6 \pm 3.0$ | $509 \pm 43$ | $2.2 \pm 0.4$ |
|  | Late | MTL | $568 \pm 51$ | $8.4 \pm 3.5$ | $516 \pm 40$ | $4.0 \pm 1.1$ |
|  | Boundary | MTL | $543 \pm 27$ | $5.6 \pm 1.4$ | $509 \pm 19$ | $4.4 \pm 1.8$ |
|  | Fused | MTL | $552 \pm 33$ | $3.2 \pm 2.0$ | $541 \pm 84$ | $3.4 \pm 1.3$ |

**Table A2.** MDE of all six experiments. Every experiment setup is trained five times with different weight initialization and train-validation-split and then averaged. The  columns show the number of images for which falsely no front was detected out of the 122 test samples.

## Appendix B:  Calving front labels

The other six glacier sites in this section are displayed complementary to Fig 4. The corresponding calving fronts of the different timesteps are colored with a gradient from bright for the past and dark red for the most recent fronts.

## Appendix C:  Temporal distribution of zone label

Figure C1 show the temporal distribution of the zone label. JAK and COL, the two glaciers in the northern hemisphere, show an increase in ocean area with a decreasing glacier area. In particular, the set of images of the glaciers on the AP has samples 355 with large areas with no available information. Low partial coverage by the radar swath causes prominent peaks of the no

| Model | Experiment | Modality | Precision↑ | Recall↑ | F1↑ | IoU↑ |
|---|---|---|---|---|---|---|
| Gourmelon et al. | Zone | STL | $84.2 \pm 0.5$ | $79.6 \pm 0.9$ | $80.1 \pm 0.5$ | $69.7 \pm 0.6$ |
| nnU-Net | Front | STL | - | - | - | - |
| | Zone | STL | $81.2 \pm 2.4$ | $71.7 \pm 3.2$ | $71.9 \pm 3.7$ | $62.4 \pm 3.5$ |
| | Early | MTL | $87.4 \pm 0.4$ | $80.9 \pm 0.7$ | $81.6 \pm 0.6$ | $72.6 \pm 0.9$ |
| | Late | MTL | $86.4 \pm 0.4$ | $79.9 \pm 0.6$ | $80.7 \pm 0.7$ | $71.1 \pm 0.7$ |
| | Boundary | MTL | $87.5 \pm 0.3$ | $80.8 \pm 0.4$ | $81.7 \pm 0.5$ | $72.6 \pm 0.4$ |
| | Fused | MTL | $87.0 \pm 0.2$ | $79.1 \pm 1.7$ | $80.7 \pm 0.1$ | $70.8 \pm 1.8$ |

**Table A3.** Segmentation metrics of all six experiments. Every experiment setup is trained five times with different weight initialization and train-validation-split. The metric results of each run are then averaged.

| Satellite: | S1 | ENVISAT | ERS | PALSAR | TDX | all |
|---|---|---|---|---|---|---|
| winter | $2529 \pm 1719\,\mathrm{m}$ | $241 \pm 101\,\mathrm{m}$ | - | - | $160 \pm 125\,\mathrm{m}$ | $818 \pm 1389\,\mathrm{m}$ |
| summer | $798 \pm 1442\,\mathrm{m}$ | $122 \pm 61\,\mathrm{m}$ | $83 \pm 39\,\mathrm{m}$ | $135 \pm 68\,\mathrm{m}$ | $197 \pm 213\,\mathrm{m}$ | $307 \pm 730\,\mathrm{m}$ |

**Table A4.** MDE of the fused label experiments grouped by sensor and season.

information available class. JAK's area distribution shows a repetitive structure of increasing and decreasing glacier area. This pattern represents seasonal changes.

| Hyperparameter | Gourmelon et al. (2022) | nnU-Net (Isensee et al., 2021) |
|---|---|---|
| Number of convolutional layers | 10 | 34 |
| Pooling | Max-pooling | Strided convolution |
| Activation-function | ReLU | Leaky ReLU |
| Patch size | 256x256 | 1280x1024 |
| Batch size | 16 | 2 |
| Optimizer | cyclic learning rate scheduler (Smith, 2017) in combination with the Adam optimizer (Kingma and Ba, 2014) | SGD with a Nesterov momentum of 0.99, and a weight decay of 3e-5 |
| Initial learning-rate | $4 \cdot 10^{-5}$ | $1 \cdot 10^{-3}$ |
| Gradient clipping | True | False |
| Deep-supervision | False | True |
| Loss-function | Combination of Dice and Cross-entropy | Combination of Dice and Cross-entropy |

**Table A5.** Comparison of the U-Net training setup and hyperparameters between Gourmelon et al. (2022) and the nnU-Net (Isensee et al., 2021).

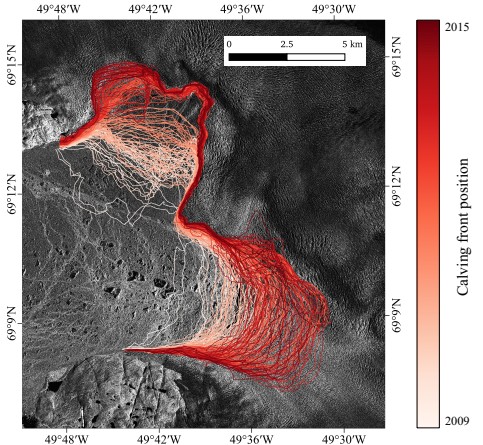

**Figure B1.** Jakobshavn Isbrae Glacier from 1995 to 2014.

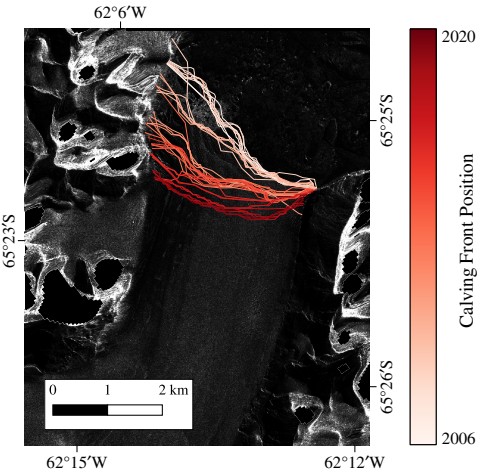

**Figure B2.** Mapple Glacier from 2006 to 2020.

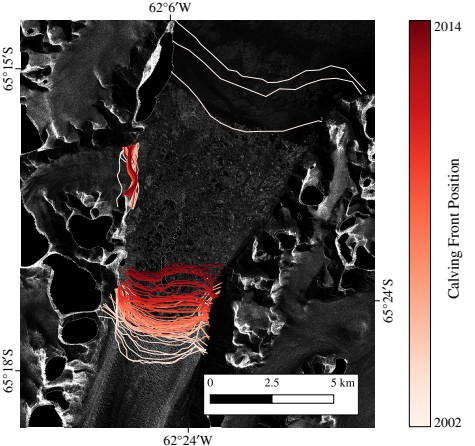

**Figure B3.** Crane Glacier from 2002 to 2014.

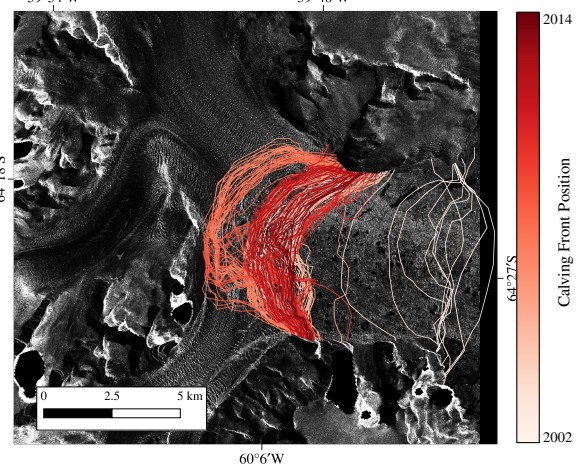

**Figure B4.** DBE Glacier from 1995 to 2014.

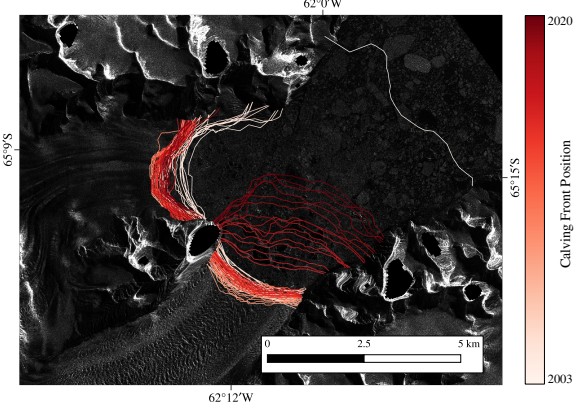

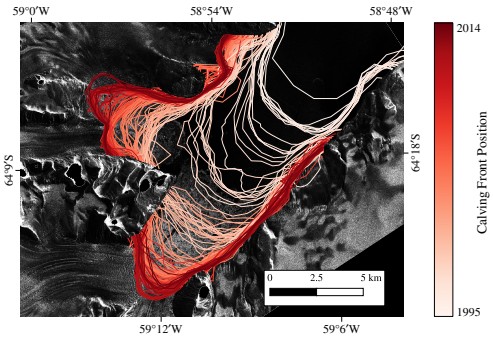

**Figure B6.** Sjögren-Inlet Glacier from 1995 to 2014.

**Figure B5.** Jorum Glacier from 2003 to 2020.

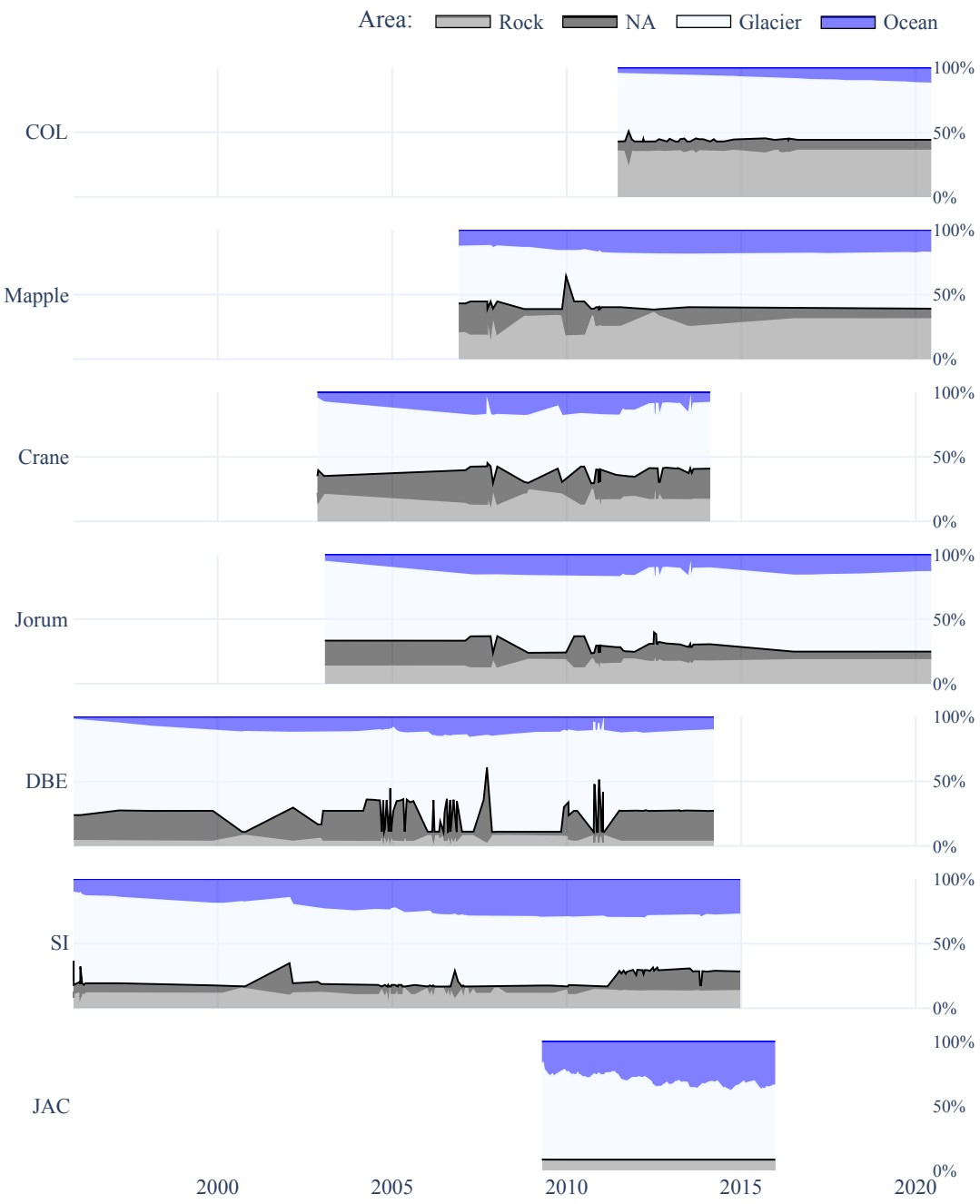

**Figure C1.** Changes of the class distributions in the zone labels of each glacier over time. The ocean is coloured in blue, the glacier area in white, the rock in grey and the area with no available information (NA) in black. There is no radar reflection in the NA area due to terrain elevation causing shadows or due to limited coverage by the radar swath.

## Code and data availability

The code is fundamentally based on the nnU-Net by Isensee et al. (2021), which is also publicly available (https://github.com/MIC-DKFZ/nnUNet). We modified the neural network architecture and corresponding pre- and post-processing. The modified version of the nnU-Net and our evaluation and visualization scripts is available on GitHub (https://github.com/ho11laqe/nnUNet_calvingfront_detection) and a Demo-version is provided on HuggingFace (https://huggingface.co/spaces/ho11laqe/nnUNet_calvingfront_detection). The calving front predictions of the test set are available on Zenodo (https://zenodo.org/record/8379954) as well as the pre-trained model (https://zenodo.org/record/7837300#.ZFuQAdIzbUA). The dataset is provided by Gourmelon et al. (2022) (https://doi.org/10.1594/PANGAEA.940950).

## Author contribution

OH and NG were responsible for conceptualizing the study. OH developed and implemented the methodology and software for the experiments. NG and TS provided the dataset and a benchmark score. NG and VC supervised the study and reviewed and edited the manuscript. JF acquired financial support. OH created visualizations and prepared the manuscript with contributions from all co-authors. JF, VC, and NG reviewed and edited the manuscript.

## Competing interests

At least one of the (co-)authors is a member of the editorial board of The Cryosphere.

## Acknowledgments

J. J. Fürst has received funding from the European Union's Horizon 2020 research and innovation programme via the European Research Council (ERC) as a Starting Grant (StG) under grant agreement No. 948290. T. Seehaus was supported by the ESA Living Planet Fellowship project "MIT-AP". We acknowledge the free provision of the SAR data via various proposals from ESA, ASF, and DLR. The authors gratefully acknowledge the scientific support and HPC resources provided by the Erlangen National High-Performance Computing Center (NHR@FAU) of the Friedrich-Alexander-Universität Erlangen-Nürnberg (FAU). The hardware is funded by the German Research Foundation (DFG). We acknowledge financial support by Deutsche Forschungsgemeinschaft and Friedrich-Alexander-Universität Erlangen-Nürnberg within the funding programme "Open Access Publication Funding". This research has been supported by the Friedrich-Alexander-Universität Erlangen-Nürnberg (Emerging Fields Initiative "Tapping the Potential of Earth Observation") and the STAEDTLER Foundation. The authors thank the Elitenetzwerk Bayern for financial support under the International Doctorate Program "Measuring and modelling mountain glaciers and ice caps in a changing climate ($M^3OCCA$)".

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
