# Peer review of "Out-of-the-box calving front detection method using deep learning"

_The Cryosphere, 2023_

## Referee Comment (RC1)

**Review on Herrmann et al 2023: Out-of-the-box calving front detection method using deep learning**

Herrmann et al. use a nnU-Net to extract the calving front of marine-terminating glaciers. To train the network they used a benchmark dataset published by Gourmelon et al. 2022 including images from six different radar sensors for seven marine-terminating glaciers. They perform six different experimental set-ups to test the best label and network set-up to achieve the highest accuracy compared to manual delineated fronts. The manuscript contains a large variety of figures. The visualizations are appealing and help to understand the study design but sometimes more profound explanations would be desirable (especially in the discussion). The manuscript discusses some interesting experiments on model set-up, labels and effect of seasonality and sensor type for the accuracy of glacier front detection. This is interesting for DL-model developers and remote sensing scientists but I would expect more added value for glaciologists as well (especially when publishing in The Cryosphere). For example, this could include the out-of-the box usage of your repository for own radar images to generate time-series for an area of interest for glaciological studies (maybe this is already possible but I cannot access the GitLab link provided in the manuscript).

**Some further comments:**

- So far, the results & discussion section is a bit difficult to follow and very short. Think about providing sub-headers and providing a more in-depth discussion.
- Addressing the seasonal effects on accuracy is very interesting. But a more sophisticated analysis would be required. In Figure 10 it looks like that the sensor type influences the accuracy much more than the season. Therefore, it would be interesting to know the sensor type for the scenes considered for winter and summer. Especially for Columbia glacier as only TSX and S1 scenes were available and for each sensor the MDE is very different.
- Provide an analysis on the effect of different spatial resolutions on the accuracy of the front extraction. Also introduce the different spatial resolutions and polarizations of all sensors as not every reader might be familiar with that.
- The conclusion is a bit unstructured and could be much stronger. Emphasize the identified best practice approach and give a better outlook for your study and applications in glaciology.

**Line specific comments:**

L16: The GitLab repository is only accessible for people affiliated to the Friedrich-Alexander-University. Please publish your scripts on a platform easily accessible for everyone.

L45: "Most of these methods use optical imagery": Three out of four cited references here use SAR imagery. Please provide proper reference for your statement.

L66: What means "visual preparation". I would assume a benchmark dataset can be used right away.

L76: Please do not only state that many methods are based on the U-Net from Ronneberger but also provide references for these studies/methods and mention that they mostly used modified versions of the original U-Net (e.g. Loebel et al. 2022, Mohajerani et al. 2019, Zhang et al. 2019 etc.)

L98: In the study of Heidler et al. 2021 the digital elevation model was used during training for different experiments. A further development of the HED-UNet for a circum-Antarctic approach (Baumhoer et al. 2023) uses indeed a DEM in the post-processing.

L145ff: If the network itself decides on batch and patch size I wonder how big the effect on the accuracy is. For example, a smaller patch size would provide less spatial context and probably

decrease the accuracy. Did you do any experiments on the effect of batch and patch size on the model accuracy? Furthermore, please state on what infrastructure (GPU type, RAM etc.) your model was trained and what patch and batch size was finally used.

L225: Maybe add a sentence why you propose that the fused label approach is the best even though the boundary label approach has better accuracies.

L245: Dry snow is penetrated by active microwave sensors and the ground can be the major source of the backscatter signal (depending on snowpack thickness and wavelength). Wet snow can be detected by SAR sensors. Hence, is snow cover really the reason for less accurate front predictions in winter? Additionally, I would assume that surface melt in summer and sea ice in winter make the front prediction more difficult. Did you investigate this further?

L228: Why did you decide to use the ensemble prediction results for the final evaluation instead of doing this evaluation for the most feasible method you identified prior to that (fused label approach)? Wouldn't it be more likely for users to use this 'best' method instead of the ensemble approach requiring much more computational power and effort?

L248: You can not directly compare the accuracies for both sensors. For ERS, Envisat and Palsar you only have data for Mapple glacier which seems to have a front that is better captured by the nn-UNet. The accuracy comparison for different sensors is very interesting especially regarding spatial resolution and polarization but this would require a more sophisticated comparison.

L250: I would recommend to investigate further why the results are worse for Sentinel-1. Maybe provide some plots in the appendix on these in-accurate results plotted over the Sentinel-1 image to see the sea ice conditions at that time.

Figure 10: Please properly label the X-axis for the logarithmic scale. Consider splitting the figure in two or insert a little gap between the seasonal analysis and the comparison of accuracy for different satellite sensors. For ERS, Envisat and Palsar you need to mention that this data is for Mapple only.

Figure 11c: The yellow front seems to be a circle. One part of the prediction fits very well whereas the other is completely off. How is this considered in the accuracy estimation?

L269: Rather "increase" model performance I guess?! To which sensor do you refer when talk about low resolution imagery?

**Literature mentioned within this review:**

Baumhoer, C. A., Dietz, A. J., Heidler, K., and Kuenzer, C.: IceLines – A new data set of Antarctic ice shelf front positions, Sci Data, 10, 138, https://doi.org/10.1038/s41597-023-02045-x, 2023.

Loebel, E., Scheinert, M., Horwath, M., Heidler, K., Christmann, J., Phan, L. D., Humbert, A., and Zhu, X. X.: Extracting glacier calving fronts by deep learning: the benefit of multi-spectral, topographic and textural input features, IEEE Transactions on Geoscience and Remote Sensing, 1–1, https://doi.org/10.1109/TGRS.2022.3208454, 2022.

Mohajerani, Y., Wood, M., Velicogna, I., and Rignot, E.: Detection of Glacier Calving Margins with Convolutional Neural Networks: A Case Study, Remote Sensing, 11, 1–13, https://doi.org/10.3390/rs11010074, 2019.

Zhang, E., Liu, L., and Huang, L.: Automatically delineating the calving front of Jakobshavn Isbræ from multitemporal TerraSAR-X images: a deep learning approach, The Cryosphere, 13, 1729–1741, https://doi.org/10.5194/tc-13-1729-2019, 2019.

---

## Author Comment (AC1)

First of all, we want to thank the reviewers for their constructive comments on our manuscript. The following pages contain a list of reviewer comments followed by our replies. The comments are sequentially numbered and associated with the corresponding reviewer. The document contains active cross-references between the replies and the modifications in the main text. The black label references the original reviewer's comment within the manuscript. For instance, the reference **1.3**, refers to the response from the third comment of the first reviewer, the reference **A2/C2/D2 (3.5)**, which is typeset in the manuscript margin, refers to the second addition/change/deletion stemming from the fifth comment of the third reviewer.

**Comment of the Editor**

*Some more detailed remarks from my side (that can be addressed during the revision process) are that when a new method is presented I think it is good practice to benchmark it to (some of the) existing methods. Otherwise, it might be difficult for the reader to effectively assess the (dis)advantage of the different methods and/or choose the optimal method for calving front detection.*

We sincerely thank the editor for the concern. A comparison between existing methods is highly needed. However, a problem arises: to compare the performance of different deep learning models, they need to be trained on the same data, tested on the same test set, and the same metric needs to be used for evaluation. Existing models have been trained and tested on different datasets (partly also different metrics were used). Hence, if we simply compare the results, we do not know whether the difference arises from the better-adapted model architecture, from a more representative train set, or from an easier test set. For this reason, Gourmelon et al. (2022) published a so-called benchmark dataset that is used for comparing the performance of different deep learning models. Our method is trained and tested on this benchmark and compared to the baseline (also presented in Gourmelon et al. (2022)) using the introduced metric. Re-training and testing other existing calving front extraction models on the benchmark is out of the scope of this study. We are, however, preparing such an inter-comparison project at the moment and hope to submit it soon.

**Comments of the 1st Reviewer:**

**1.1** *So far, the results & discussion section is a bit difficult to follow and very short. Think about providing sub-headers and providing a more in-depth discussion.*

▷ Thank you for your suggestion. We added sub-headers (additions 10 and 11) and will include Fig. 11, which gives insights into the influence of resolution on the calving front prediction (comment 1.3). We added a more in-depth discussion on the seasonal impact (changes 13 and 14).

**1.2** *Addressing the seasonal effects on accuracy is very interesting. But a more sophisticated analysis would be required. In Figure 10 it looks like that the sensor type influences the accuracy much more than the season. Therefore, it would be interesting to know the sensor type for the scenes considered for winter and summer. Especially for Columbia glacier as only TSX and S1 scenes were available and for each sensor the MDE is very different.*

▷ Thank you for raising this interesting discussion. We include Table A4 in the Appendix and refer to it in addition 16. The Table shows the Mean Distance Error (MDE) grouped by satellite and seasons. The influence of the sensor on the MDE can also be seen in Fig. 12. Comparing the rows Mapple_TDX with Mapple_S1 and Columbia_TDX with Columbia_S1 gives the differences introduced by sensors without the influence of the test site. As you state, the sensor type influences the MDE much more than the season.

**1.3** *Provide an analysis on the effect of different spatial resolutions on the accuracy of the front extraction. Also introduce the different spatial resolutions and polarizations of all sensors as not every reader might be familiar with that.*

▸ Thank you for raising this point. We now introduce the different spatial resolutions in addition 4. We provide Fig. 11 and add analysis to the manuscript (addition 17). We added the resolution of each satellite in the caption. Unfortunately, the used benchmark dataset does not provide information about the polarization of the sensor for each image.

**1.4** *The conclusion is a bit unstructured and could be much stronger. Emphasize the identified best practice approach and give a better outlook for your study and applications in glaciology.*

▸ Thank you very much for your suggestions. We revised the conclusion by emphasizing the best practice approach and added a better outlook of the applications for the cryosphere community (change 17 and additions 22, 24 and 25)

**1.5** *L16: The GitLab repository is only accessible for people affiliated to the Friedrich-AlexanderUniversity. Please publish your scripts on a platform easily accessible for everyone.*

▸ Thank you for raising this issue. We uploaded the code on GitHub (https://github.com/ho11laqe/nnUNet_calvingfront_detection) and HuggingFace (https://huggingface.co/spaces/ho11laqe/nnUNet_calvingfront_detection). HuggingFace provides computing resources to execute programs directly via the web portal. This should be easily accessible to everyone (changes 1 and 18).

**1.6** *L45: "Most of these methods use optical imagery": Three out of four cited references here use SAR imagery. Please provide proper reference for your statement.*

▸ Thank you for pinpointing this inaccuracy. We changed the statement (changes 2 and 3 and addition 1).

**1.7** *L66: What means "visual preparation". I would assume a benchmark dataset can be used right away.*

▸ Thank you for pointing out this confusion. The benchmark can be used right away. With the term "visual preparation", we meant visualizations that provide an overview of the temporal and spatial distribution of the dataset. To be more precise, the term "visual preparation" refers to Fig. 4, 5, and Appendix C. We clarified this phrase in the revised version of the paper (change 4).

**1.8** *L76: Please do not only state that many methods are based on the U-Net from Ronneberger but also provide references for these studies/methods and mention that they mostly used modified versions of the original U-Net (e.g. Loebel et al. 2022, Mohajerani et al. 2019, Zhang et al. 2019 etc.)*

▸ Thank you for this suggestion. We added the references to the revised version of the paper and mentioned that they mostly used modified versions of the original U-Net (addition 2).

**1.9** *L98: In the study of Heidler et al. 2021 the digital elevation model was used during training for different experiments. A further development of the HED-UNet for a circum-Antarctic approach (Baumhoer et al. 2023) uses indeed a DEM in the post-processing.*

▸ Thank you for pointing out this inaccuracy. We edited this segment of the text and added the article from Baumhoer et al. (change 5 and addition 3).

**1.10** *L145ff: If the network itself decides on batch and patch size I wonder how big the effect on the accuracy is. For example, a smaller patch size would provide less spatial context and probably decrease the accuracy. Did you do any experiments on the effect of batch and patch size on the model accuracy? Furthermore, please state on what infrastructure (GPU type, RAM etc.) your model was trained and what patch and batch size was finally used.*

▸ This is an interesting point. We did not evaluate the effect of patch and batch size on the model. Our goal was to evaluate the nnU-Net's ability to segment the Synthetic Aperture Radar (SAR) dataset without modifications by us. The authors of the

nnU-Net also suspect that a large patch size has a positive effect on accuracy. The nnU-Net makes the most of the available GPU memory for best performance. For reproducibility, we added a subsection about the experimental setup in the manuscript (addition 8). We trained the nnU-Net on an NVIDIA RTX 3080 with 12GB memory. This allowed a patch size of 1280x1024 for the experiments that had only one label. The experiments with more than one label have a patch size of 1024x896. For all experiments, the batch size is 2.

**1.11** *L225: Maybe add a sentence why you propose that the fused label approach is the best even though the boundary label approach has better accuracies.*

▶ Thank you for pointing out the confusing sentences. The mean accuracy of the boundary experiment might be better after our training run, but the difference is insignificant. We evaluated a T-test in table A1 to reveal significant differences. Therefore we choose this training approach because it uses the smallest amount of parameters and does not need a change of architecture as in the Multi-Task-Learning (MTL) experiments. We edited the sentence to make clear why we chose the fused label approach (change 12).

**1.12** *L245: Dry snow is penetrated by active microwave sensors and the ground can be the major source of the backscatter signal (depending on snowpack thickness and wavelength). Wet snow can be detected by SAR sensors. Hence, is snow cover really the reason for less accurate front predictions in winter? Additionally, I would assume that surface melt in summer and sea ice in winter make the front prediction more difficult. Did you investigate this further?*

▶ We understand the reviewer's concerns. Not only the snow cover and its properties, also the surface melt, and sea ice make the front prediction more difficult. In particular, the ice melange is certainly causing some issues when detecting the calving front. Even for an experienced human mapper, it can be very challenging to distinguish between the actual glacier tongue and the ice melange. We adjusted the wording of the respective section accordingly (changes 13 and 14).

**1.13** *L228: Why did you decide to use the ensemble prediction results for the final evaluation instead of doing this evaluation for the most feasible method you identified prior to that (fused label approach)? Wouldn't it be more likely for users to use this 'best' method instead of the ensemble approach requiring much more computational power and effort?*

▶ Thank you for pointing out the confusing explanation. The final evaluation does concern only the most feasible method. The ensemble consists of the five models that are trained during the five-fold cross-validation. Every model of this ensemble is trained on the fused label, but each one is trained on a different train-validation split of the training set. We added information to the manuscript to clarify this confusion (additions 12, 13 and 15).

**1.14** *L248: You can not directly compare the accuracies for both sensors. For ERS, Envisat and Palsar you only have data for Mapple glacier which seems to have a front that is better captured by the nnUNet. The accuracy comparison for different sensors is very interesting especially regarding spatial resolution and polarization but this would require a more sophisticated comparison.*

▶ Thank you for pointing this out. We clarified that ERS, ENVISAT, and PALSAR only capture Mapple Glacier (change 15 and addition 18). Moreover, we split the satellites into glaciers as well in Fig. 12. We refer to our answer on the comment 1.3 where we group the results by resolution. Unfortunately, the dataset does not provide information about the polarisation of the sensor.

**1.15** *L250: I would recommend to investigate further why the results are worse for Sentinel-1. Maybe provide some plots in the appendix on these in-accurate results plotted over the Sentinel-1 image to see the sea ice conditions at that time.*

▶ Thank you for this suggestion. It seems like the nnU-Net is more vulnerable to being distracted by ice melange in the images of Columbia taken by Sentinel-1 (S1), but there are also predictions where the calving front is in the middle of the ice-free ocean. We suspect that the complex Columbia calving front requires high resolution for accurate detection, due to the

lower MDE of the TDX images of Columbia or more training samples of complex calving fronts with low resolution. We stick to our resilient conclusion that our method performs insufficient on the group of images of Columbia captured by S1 (addition 20). Further experiments that could verify our hypothesis that for the Columbia Glacier, a higher resolution than that of S1 is needed are out of the scope of this manuscript. However, we provide all predictions of the test set on Zenodo (https://zenodo.org/record/7915740#.ZFtS3dIzZhE). We added the link to the predictions in the manuscript (addition 21).

**1.16** *Figure 10: Please properly label the X-axis for the logarithmic scale. Consider splitting the figure in two or insert a little gap between the seasonal analysis and the comparison of accuracy for different satellite sensors. For ERS, Envisat and Palsar you need to mention that this data is for Mapple only.*

▸ Thank you very much for the helpful comments. We split the results plot into two. One contains the glacier and season comparison and the other contains the satellite comparison. We changed the labels of the X-axis and stated that ERS, Envisat and PALSAR only capture the Mapple glacier (Fig. 10, Fig. 12).

**1.17** *Figure 11c: The yellow front seems to be a circle. One part of the prediction fits very well whereas the other is completely off. How is this considered in the accuracy estimation?*

▸ This raises an interesting point. For each pixel in the predicted front, the distance to the closest pixel in the labelled front is computed, and for every pixel in the labelled front, the distance to the closest pixel in the predicted front is computed. The mean of all the distances is taken as the MDE. The MDE of this particular calving front prediction is $4221 \pm 5026\,\mathrm{m}$ (addition 19). The metric does not take into account geometric impossibilities like a calving front forming a circle. However, one side of a circle will still be far from the ground truth, and therefore, it will still negatively impact the MDE. We stick to this metric to enable comparability with the baseline presented by Gourmelon et al. (2022).

**1.18** *L269: Rather "increase" model performance I guess?! To which sensor do you refer when talk about low resolution imagery?*

▸ Thank you for this suggestion. We changed the word "benefits" to "increase" (change 16). We mean the S1 sensor with $20\,\mathrm{m}$ per pixel resolution (addition 23).

**Comments of the 2nd Reviewer:**

**2.1** *The author applies a nnU-Net and improves the error from 753 ± 76 m by Gourmelon et al. (2022) to 541 ± 84 m. Still, 541 meters is a relatively large error for terminus delineation, compared with previous studies that have errors ranging from 33 to 108 meters (Zhang et al., 2019; Cheng et al., 2021; Baumhoer et al., 2019). Given the relatively large margin of error in the terminus products produced by this method, it might be challenging to conduct scientific research that relies on them. Also, providing more termini that go beyond the benchmark dataset might be more beneficial to the glaciology community, but I understand it might be out of the scope of this study.*

▶ We sincerely thank the reviewer for her/his concern. First of all, we would like to state that this manuscript is a methodology paper, which aims to improve the method used to extract calving front positions from SAR imagery (semi-)automatically. Therefore, as the reviewer already kindly states, it is out of the scope of this study to provide additional termini outside the benchmark dataset. The reviewer also correctly states that researchers should not directly rely on terminus products provided by our method. Our method should be used semi-automatically, like, e.g., CALFIN (Cheng et al., 2021), where a manual verification of each output is done with the original image (only this manual verification leads to a mean distance error < 100 m). We do not claim that our method perfectly extracts all calving fronts, and we now state more clearly in our conclusion that the output needs manual verification (see addition 22).

However, we object the direct comparison of our study results with the result from previous studies. To correctly assess the performance difference between two deep learning models, the models need to be trained on the same training data, tested on the same test set and evaluated with the same metrics. Otherwise, it is not possible to discern where the performance difference stems from.

Here we would like to highlight some reasons in particular that prohibit a comparison with previous studies:

(1) In some studies, an in-sample test set is used (e.g., Zhang et al. (2019), Hartmann et al. (2021) and partly also Cheng et al. (2021)), i.e., the test set contains images of glaciers that are also included in the train-set, but at different time points. A test on an in-sample test set is easier, as the model does not need to generalize to a new environment.

(2) In comparison to our test set, some other test sets are not as challenging (e.g., Zhang et al. (2021) - Helheim Glacier features only one calving front with a simple geometry) or rather small (Baumhoer et al. (2019) - 8 samples all from the same month).

(3) Other test sets, in turn, have only high-resolution imagery (e.g., Zhang et al. (2019)), which we at least hypothesize helps the automatic calving front extraction with deep learning methods.

(4) Baumhoer et al. (2019) delineate the complete ice shelf/ice sheet coastline, not a glacier calving front bounded by bedrock. Hence, a direct comparison can not be drawn.

(5) Our test set, in contrast to others (e.g., Baumhoer et al. (2019) and Zhang et al. (2019)), features multiple sensors introducing more variability and, therefore, making it more challenging for deep learning models.

(6) The comparison to the averaged performance value of Cheng et al. (2021) lacks in validity two-fold: 1. The average performance is in big parts calculated over results from optical imagery. CALFIN's results on subsets of the test set that only include SAR imagery are 115.24 m on Zhang et al. (2019)'s test set and 330.63 m on Baumhoer et al. (2019)'s test set. 2. The performance metrics are calculated after manual verification. Thus, outliers that would lead to a higher MDE are not included anymore.

(7) In general, two metric values are important: the MDE and the number of samples with no predicted front. A trade-off between the two has to be made. For difficult samples, is it desirable to predict a front with a high MDE, or is it better to predict no front at all? For example, Cheng et al. (2021) sort out unreliable predictions by confidence scores and by hand and, thus, receive a lower MDE but a higher number of samples with no predicted front (on the Zhang et al. (2019) test set $4 \in 12$ results are filtered out, while for our method only $5 \in 122$ predictions show no front).

To improve the comparability of deep learning models for calving front extraction from SAR imagery, Gourmelon et al. (2022) published a so-called benchmark dataset with a corresponding evaluation metric. We only compare to the baseline models provided in (Gourmelon et al., 2022) and improve over the baseline's performance. A re-training and re-evaluation of previous calving front extraction models is out of the scope of this study. However, we are preparing such an intercomparison study so that this study's method can be compared with previous studies in a rigorously correct manner.

**2.2** *One key advantage of using no new U-Net is that the framework can take the fingerprint of the dataset and automatically configures the framework (adjust the hyperparameters). However, it is not clear to me which hyperparameters are being adjusted automatically, how they are adjusted, and what is the final choice of these hyperparameters. If we adopt the hyperparameters from nnU-Net and apply them to standard U-Net, would there be no difference between nnU-Net and U-Net? If that is the case, I suppose it is important to clearly demonstrate how to automatically adjust the hyperparameters and how these newly derived hyperparameters improve the results.*

▸ Thank you for pointing out this incomplete description of the nnU-Net. A lot of hyperparameters are fixed by the authors of the nnU-Net. This includes the optimizer (Stochastic gradient descent (SGD)) with an initial learning rate of 0.01, a Nesterov momentum of 0.99, and a weight decay of 3e-5. One epoch is defined as 250 iterations. The batch and patch size of one iteration depends on the available GPU memory. In our case, that leads to a patch size of 1280x1024 for the experiments with only one label. The experiments with more than one label have a patch size of 1024x896. For all experiments, the batch size is 2. The number of layers of the neural network depends on the patch size. The authors introduce building blocks containing layers for the nnU-Net (see Fig. 1). Each block reduces the spatial dimensions. Blocks are added to the network until the feature maps have a shape of 4x4 pixels. The building blocks of the nnU-Net contain Instance Normalization, which the vanilla U-Net didn't, they use Leaky ReLU instead of ReLU as an activation function, and they use a stride of [2,2] instead of max-pooling to reduce spatial dimensions.

In addition, to the hyperparameters, the nnU-Net has sophisticated training techniques, that are not included in the vanilla U-Net. The nnU-Net uses Deep Supervision. This technique avoids vanishing gradients in deep neural networks (containing many layers). An additional output of each decoder block is compared to a down-scaled version of the target label. The error is multiplied by a weight and added to the final loss. The weight increases towards the final layer.

Besides the architectural changes, there are also changes in the training and inference procedure. In the work of Ronneberger et al. (2015), they already used data augmentation (shift, rotation, random elastic deformation, and gray value variations). The nnU-Net framework extends the list of augmentation techniques (see Table 1). Another technique for the inference procedure sets the vanilla U-Net and the nnU-Net framework apart. For more robust predictions, a sliding window with half-patch size overlap and Gaussian patch centre weighting is used for inference. Additionally, each patch is rotated three times (90°, 180°, 270°) and the predictions of each rotation are fused to a more robust prediction.

To summarize, the nnU-Net of Isensee et al. (2021) is a framework around the U-Net architecture. In addition to providing good default values for hyperparameters and rules to adapt hyperparameters to the dataset, it contains many established deep-learning techniques for training and inference that make predictions more robust.

We added a new subsection to the manuscript, which provides the hyperparameters of the nnU-Net generated and information about the experimental setup (addition 8). We made changes in the Background (change 7).

The improvement of the nnU-Net framework techniques over another version of the U-Net is given by the comparison to the work of Gourmelon et al. (2022) (see Fig. 9, Fig. 10, and Fig. 12).

**2.3** *While the author emphasizes that the method is an out-of-the-box application, it would be beneficial for the readers to have a clearer understanding of the steps taken by the author to ensure its out-of-the-box applicability. Providing further details on this aspect could enhance the credibility of the method's out-of-the-box applicability claim and contribute to its broader adoption in the research community.*

▸ Thank you very much for the helpful comment. This was actually a misunderstanding. With "out-of-the-box" we initially did not want to imply that a reader can use our pre-trained model to detect calving fronts in their satellite images with little effort. Instead, we meant that we use the unmodified (out-of-the-box) nnU-Net on the dataset of Gourmelon et al. (2022). We now see that this was misleading. Regardless of the interpretation of the title, we want to make our method easily accessible - to make our trained model *itself* useable out-of-the-box. We provide two ways to use our pre-trained

[Figure]

**Figure 1.** Illustration of the encoder and decoder blocks that make up the architecture of the nnU-Net. The encoder and the decoder contain multiple blocks.

| Augmentation | Parameters | Probability |
|---|---|---|
| Elastic Deformation | | 0% |
| Rotation | x=[-3.14, 3.14] | 20% |
| Scaling | factor=[0.7, 1.4] | 20% |
| Random Crop | | 0% |
| Gaussian Noise | add noise | 10% |
| Gaussian Blur | add Blur sigma=[0.5, 1] | 20% |
| Brightness Multiplication | range=[0.75, 1.25] | 15% |
| Contrast Augmentation | | 15% |
| Low Resolution | zoom=[0.5, 1] | 25% |
| Gamma | gamma=[0.7, 1.5], | 30% |
| Mirroring | x- and y-axis | 50% |

**Table 1.** List of augmentations used by the nnU-Net Framework. The table includes the parameters of the augmentations and the probabilities of their usage.

nnU-Net for calving front detection. One is via GitHub (https://github.com/ho11laqe/nnUNet_calvingfront_detection) with a short instruction for installation and application on SAR images. An even simpler way is provided with HuggingFace (https://huggingface.co/spaces/ho11laqe/nnUNet_calvingfront_detection). The code is executed directly on the website, so the

reader does not have to install any Python packages. The drawback is that the code runs much slower than on a PC with a GPU, so it is suited only for a small number of SAR images. We added the links to the according paragraphs in the manuscript (changes 1 and 18).

**2.4** *Line 16: The scripts are not publicly available. As the author emphasizes that the method is out-of-the-box, it would be more beneficial if the codes are publicly available.*

▸ Thank you for pointing this out. We uploaded the scripts to GitHub (https://github.com/ho11laqe/nnUNet_calvingfront_detection) and HuggingFace (https://huggingface.co/spaces/ho11laqe/nnUNet_calvingfront_detection) to make it publicly available (change 1).

**2.5** *Line 66: What is the visual preparation? Why is the visual preparation necessary as the dataset from Gourmelon et al. (2022) is a benchmark dataset?*

▸ Thank you for pointing out this inaccurate term. No visual preparations of the dataset are necessary. With the term "visual preparation", we meant visualizations that provide an overview of the temporal and spatial distribution of the dataset. To be more precise, the term "visual preparation" refers to Fig. 4, 5, and Appendix C. We clarified this phrase in the revised version of the paper (change 4).

**2.6** *Line 115: Are sentences about the labels here and the sentences on Line 50 repeated?*

▸ Thank you for pointing this out. The content of these lines does repeat. After considering shortening the sentence in the introduction, we keep the composition of the labels in both chapters because they occur in two separate chapters and transport important information to the reader.

**2.7** *Figure 3: For Crane, what is the black dots in the middle of the ocean?*

▸ Thank you for your interest. The black dots in the ocean zone of Crane are areas with no data available due to a small artefact at the former calving front of Crane Glacier in the DEM (Cook et al., 2012) used for the orthorectification of the SAR data. We added the information to Fig. 3.

**2.8** *Line 120: Are there any procedures to deal with the imbalance? It might be helpful to increase the weight of the positive pixel loss in the loss function.*

▸ The reviewer raises an excellent point. There is a great imbalance between front pixels and background pixels. One technique of the nnU-Net to reduce the class imbalance is foreground oversampling. It is ensured that (at least) one third of the patches for training are guaranteed to contain a foreground class. In our case, every batch contains two patches, from which one is forced to contain calving front pixels.

Additionally, the Dice-loss is part of the final loss. The Dice-loss is defined in Eq. 1.

$$l_{dice}(y) = \frac{2|X \cap Y|}{|X| + |Y|} \tag{1}$$

Where $|X|$ represents the number of pixels assigned to one class in the prediction, and $|Y|$ is the number of pixels of a class in the target. $|X \cap Y|$ the number of pixels assigned to the same class in prediction and label. Because the number of pixels per class is included in the denominator and is computed in each mask separately, the loss is normalized so that every class is treated equally (Li et al., 2019a).

We added an explanation about how the class imbalance problem is faced in additions 5 and 6.

**2.9** *Figure 5: Please explain the abbreviations of the glacier names.*

▸ Thank you for your comment. We added an explanation of the abbreviations in the caption change 6 and in the caption of Fig. 5.

**2.10** *Section 4.1: Additional information about the method needs to be provided. It would be beneficial to include more details on the data fingerprint and how to use the fingerprint to determine the hyperparameters in this study. The following are some specific descriptions that is unclear to me:*
*What is the definition of the distribution of spacing?*
*How to use the distribution of spacing to determine the annotation, image resampling strategy, and target spacing?*
*Is the annotation equivalent to labeling the images? If so, how is that related to the distribution of the spacing?*
*What is the image resampling strategy that is used?*

▸ Thank you for this clarifying question. The distribution of spacing refers to three-dimensional (3D) medical data like Computer Tomography (CT), with different spacing between layers. Since the SAR have equal spacing, no re-sampling strategy is necessary. We now omit parts of this section since they are not relevant to our experiments (changes 8 and 9).

**2.11** *Is CT or z-score normalization used in this study? How to conduct the z-score normalization with image mean and standard deviation?*

▸ Thank you for the clarifying question. Yes, z-score normalization is used. The mean is subtracted from the pixel value and divided by the standard deviation. We clarify this in change 10.

**2.12** *Line 175: Please be more clear about the minor changes here.*

▸ Thank you for your comment. By minor changes, we mean changing the number of channels in the case of MTL. We clarified this in change 11.

**2.13** *Line 180: What is the post-processing step for the fifth experiment? How to combine the three types of output to get a single glacier terminus?*

▸ Thank you for pointing out the confusing description of the post-processing. Only the zone output is used to get the final front position. The other labels help the nnU-Net during training to adjust the weights so that they fit better to the application (addition 7).

**2.14** *Line 188: "To generate a prediction of the zones, pixels classified as front are assigned to the ocean, and the glacier zone is dilated once with a 7x7 kernel." Do the zones here represent the segmentation zone? If so, how does this lead to a glacier terminus? Similar to the fifth experiment, how to combine the two types of output to get a single glacier terminus?*

▸ Thank you for your comment. Yes, the zones in this sentence represent the segmentation zone. We added the post-processing to extract the zone segmentation and the front segmentation from the fused prediction. This way, we can compare it to the other MTL experiments. For the final evaluation, we omit the segmentation metrics (table A3) and focus on the MDE. The counterintuitive part is that we use the zone segmentations to get the front position instead of directly taking the front segmentation. There are predictions with a neighbouring glacier and ocean zone with no front prediction in between. We assign the ocean pixels that have neighbouring ocean pixels as front pixels. Therefore the front position that is extracted from the zone can differ from the front segmentation even in the fused label approach. We mentioned this in section 4.2, but we will add it also to the section Analysis of the fused label experiment (addition 14) and section Evaluation (addition 9).

**2.15** *Line 219: Why the errors of the STL approaches here are larger than the error of Gourmelon et al. (2022)?*

▸ This raises an interesting point. After adaptation, the nnU-Net has eight pooling (and un-pooling) steps, whereas Gourmelon et al. (2022) have only four pooling steps but introduce Atrous Spatial Pyramid Pooling (ASPP) in the bottleneck. One hypothesis is that ASPP works better than simply increasing the pooling steps (and, with that, the encoder blocks). However, the success of both pooling and ASPP also relies on the patch size, which is another unknown in the question of why the performance of the networks differ. There are many differences between the two models, and it is out of the scope of this study to test which subset of these many changes makes the difference. We can only give a more detailed comparison between the two methods by including Table A5.

[revised manuscript text omitted]

---

## Author Response (AR2)

**Response to the second round of revisions on "Out-of-the-box calving front detection method using deep learning" by Oskar Herrmann er al.**

First of all, we want to thank the reviewers again for their constructive comments on our revised manuscript. The following pages contain a list of reviewer comments followed by our replies. The comments are sequentially numbered and associated with the corresponding reviewer. The document contains active cross-references between the replies and the modifications in the main text. The black label references the original reviewer's comment within the manuscript. For instance, the reference **1.3**, refers to the response from the third comment of the first reviewer, the reference **A2/C2/D2 (3.5)**, which is typeset in the manuscript margin, refers to the second addition/change/deletion stemming from the fifth comment of the third reviewer.

**Comments of the 1st Reviewer:**

**1.1** *My main concern is about the impact of this study. The ultimate goal of automating terminus extraction is to provide as many as quality-controlled termini for glaciological community to conduct scientific research. I understand it might not be the scope of this study to achieve this such an ultimate goal, but I concern that a high uncertainty might undermine the impact of this paper, even though this is a methodology paper. I believe the key contribution of this study is to improve the error from 753m to 541 m, compared with Gourmelon et al. (2022). However, both errors are relatively high.*

▸ Thank you for your comment. We invite you to look at the animations uploaded to Zenodo (https://zenodo.org/record/8379954). We provided an animation of the two test sites. The Mean Distance Error (MDE) seems to be high, but the animation gives a better insight into the weaknesses and strengths of the method. You can see that in the images of the Columbia Glacier, the predicted calving front is longer than the labelled front and extends on the coastline. This has a negative influence on the MDE, but the predictions are still useful to show changes in the calving front. If the coastline is known, the misclassified part can easily be removed. In nearly all predictions of Columbia (COL) taken by TanDEM-X (TDX) the northern coastline was misclassified as a calving front. It is a difficult task to distinguish this particular coastline from the calving front because the glacier retreated so far that it transitioned from a marine-terminating glacier to a land-terminating glacier. Since the glacier front is close to the coastline, even humans find it difficult to distinguish between the two.

The median MDE is another representation of the performance of the method and gives a different impression. The median is the value that separates the lower and the higher half of the set of MDE of the test set samples. In our case, this is $153\,\mathrm{m}$, which is also shown in Fig. 10. This shows that half of the predictions have an MDE below $153\,\mathrm{m}$. The mean MDE is heavily influenced by a group of outliers, with MDE greater than $2000\,\mathrm{m}$ in contrast to the median MDE. If you would exclude these twelve outliers, the mean MDE would drop from $541\,\mathrm{m}$ to $181\,\mathrm{m}$. Similar post-processing steps were used by Zhang et al. (2021), where a complexity score is computed for each front. Predictions above a certain threshold were discarded, and additionally, predictions were discarded manually. In another work from Heidler et al. (2021) the metric is only calculated within a $2000\,\mathrm{m}$ margin of the true coastline. Davari et al. (2022b) use a tolerance belt around the ground truth.

We dispense with such techniques to reduce the MDE to maintain comparability to the baseline from Gourmelon et al. (2022) and to show the "out-of-the-box" performance of the network itself. If the model is used for calving front detection, we can recommend the aforementioned post-processing steps using the position of the coastline.

We added another point for discussion in addition 3 and updated the Zenodo link in the manuscript (changes 2 and 3).

**1.2** *2.2 In Line 66, the author mentioned that the nn-U-Net is a framework for training the U-Net and can adjusts the hyperparameter accordingly. Therefore, in the previous comments, I was wondering which hyperparameters the framework adjust and how the it is done based on the dataset automatically (accordingly). In the response, it seems to me that the framework did not adjust the hyperparameter according to the dataset. The patch size and the batch size are decided by author based on the GPU memory, which is commonly adopted when using any other deep learning methods. So, I suggest to remove the sentence about the nn-UNet here can adjusts the hyperparameters according to the dataset. With that being said, I am still wondering that why the nn-UNet here does not adjust the hyperparameters, as being a self-configuring method is the most important feature of the nn-UNet (Isensee et al., 2021).*

▸ Thank you for pointing out the confusing sentence. Our intention in line 66 was to introduce the nnU-Net in general. As you correctly point out, the self-adjusting hyperparameters play a minor role in our work, because most rule-based hyperparameters

are only used for three-dimensional (3D) segmentation with different spacings between pixels. Therefore, we rephrased the sentence in change 1 to emphasize the fixed hyperparameter over the rule-based hyperparameters.

We mentioned that we set the patch and batch size to maintain comparability between our experiments, but the initial value is generated by the nnU-Net with the early-branching architecture based on the median size of the training samples and the GPU memory. We slightly changed section 4.1, to make this clearer (see deletion 1 and addition 1.

**1.3** *2.3 If "out-of-the-box" means "unmodified", why it is important to highlight in the title that the author use the unmodified nn-UNet? In line 72, the second contribution is the out-of-the-box application of the nn-UNet. It seems a little odd to me to highlight "unmodified" application of a method.*

▸ We want to emphasise the "unmodified" use of a method in the title because we demonstrate the generalisation capabilities of the nnU-Net from the medical domain to a completely unrelated domain of "remote sensing". In addition, we modified the nnU-Net for Multi-Task-Learning (MTL) and showed that the "unmodified" architecture worked equally well with fused "modified" labels. Referring to line 72, the contribution of applying an unmodified framework to a new dataset lies in (1) installing the nnU-net, (2) converting the dataset into pseudo-3D Neuroimaging Informatics Technology Initiative (NIfTI), which is a file format used for Computer Tomography (CT) and Magnetic Resonance Imaging (MRI) scans, (3) training the nnU-Net on five folds of the dataset, which took five days on an Nvidia RTX3080, for each set of labels (fronts and zones) (4) converting the output back from pseudo-3D to further usable Portable Network Graphic (PNG) and (5) evaluating the results using MDE (see table A2) and Segmentation metrics Precision, Recall, F1-score and Intersection over Union (IoU) (see table A3). The files for the data conversion can be found in our GitHub project in the folder nnunet/dataset_conversion in the files Task501_Glacier_front.py, Task501_Glacier_reverse.py, Task502_Glacier_zone.py, Task502_Glacier_reverse.py. We don't say that this is our main contribution, but it is at least worth mentioning. From our perspective, the main contribution are points three, four and five from line 74, which refer to the implementation and evaluation of different ways to include both labels (zone and front) into the same training of the nnU-Net. The title refers to our conclusion of the comparison: all experiments that incorporate both labels perform better than the single-task experiments, emphasizing that the out-of-the-box architecture trained on the fused labels performs as well as the invasive experiments with a different architecture. We rephrased the contributions in section 1.

**1.4** *I tried HuggingFace link, however it shows error after I upload one image of a glacier and click submit. I would suggest to add additional information (e.g., image format) in the webpage or the manuscript to make the link work.*

▸ Thank you for trying it out. We are sorry that it didn't work for you. This can have multiple reasons: (1) The input image was not a PNG. Since the network was trained on PNG of the dataset, we stick to this file format. (2) The image was too large. Unfortunately, this platform provides limited computational resources, we recommend a maximum size of 200KB, which takes about 50 seconds to process. For larger files the HuggingFace project can be downloaded and run on a local PC.

We elaborated the description of the interface and included two examples
(see https://huggingface.co/spaces/ho11laqe/nnUNet_calvingfront_detection).

**1.5** *2.13 If I understand correctly, in the multi-task learning experiments, the task of delineating calving front only and delineating the whole glacier boundary are abandoned in the inference, but only help to adjust the parameters in the training progress.*

▸ Thank you for highlighting this fact. Yes, during inference the other prediction is omitted. We made it a bit clearer in addition 2.

**1.6** *Another related issue is about the fused label. If the author fuse the label of calving front into the zone label with an additional class, I think it is a single classification task instead of a multi-task method. It is the same with the original zone segmentation but with one more class representing the calving front. Also, as this fused label yields the best results, it might be better to change the conclusion throughout the manuscript regarding the multi-task-learning.*

▸ Thank you for your comment. This is a point of discussion. From a technical point of view, we fuse two different labels into one and we can use the unmodified Single-Task-Learning (STL) architecture. Nevertheless, they come from two different tasks,

and as such we refer to it as MTL since we use both labels in contrast to the true STL experiments. We show this ambiguity in Fig. 6. We try to circumvent the hard categorisation of the fused label training into STL or MTL, because of the ambiguity. The conclusion still holds true as we state that "The results show that combining both tasks increases each task's performance.". We rephrased another sentence in addition 4,

**Comments of the 2nd Reviewer:**

**2.1** *Fig 3e: For some reason the front of Jacobshavn is almost not visible.*

▸ Thank you for pointing this out. The different resolution and size of the images is the reason for the calving front to appear thinner. We dilated the front label in Fig. 3, for better visibility.

**2.2** *L 146: Would it be possible to approach Gourmelon et al. to get information about the polarizations? The person that pre-processed the images should now the details about the SAR data.*

▸ Thank you for your request. We approached the person, but unfortunately, it was not possible to receive information about the polarisation. The polarization was not logged explicitly. Retroactively researching the polarization would be out of the scope of this work.

[revised manuscript text omitted]